# Understanding potential-dependent competition between electrocatalytic dinitrogen and proton reduction reactions

Changhyeok Choi[1], Geun Ho Gu[1], Juhwan Noh[1], Hyun S. Park [2] & Yousung Jung [1✉]

A key challenge to realizing practical electrochemical $N_2$ reduction reaction (NRR) is the decrease in the NRR activity before reaching the mass-transfer limit as overpotential increases. While the hydrogen evolution reaction (HER) has been suggested to be responsible for this phenomenon, the mechanistic origin has not been clearly explained. Herein, we investigate the potential-dependent competition between NRR and HER using the constant electrode potential model and microkinetic modeling. We find that the H coverage and $N_2$ coverage crossover leads to the premature decrease of NRR activity. The coverage crossover originates from the larger charge transfer in $H^+$ adsorption than $N_2$ adsorption. The larger charge transfer in $H^+$ adsorption, which potentially leads to the coverage crossover, is a general phenomenon seen in various heterogeneous catalysts, posing a fundamental challenge to realize practical electrochemical NRR. We suggest several strategies to overcome the challenge based on the present understandings.

[1] Department of Chemical and Biomolecular Engineering (BK21 four), Korea Advanced Institute of Science and Technology (KAIST), Daejeon, Republic of Korea. [2] Center for Hydrogen and Fuel Cell Research, Korea Institute of Science and Technology (KIST), Seoul, Republic of Korea. ✉email: ysjn@kaist.ac.kr

Ammonia, the main source of nitrogen fertilizers, is one of the most produced chemicals in the world (e.g., 150 million metric tons in 2019)[1]. Ammonia has been primarily produced by the Haber–Bosch process, proceeding via the net reaction of $N_2 + 3H_2 \rightarrow 2NH_3$. To dissociate the strong $N \equiv N$ triple bond of N and shift the equilibrium towards ammonia, the Haber–Bosch process typically requires harsh conditions of ~400 °C and ~150 bar[2]. Hence, ammonia production is responsible for 1~2% of worldwide energy consumption. Also, a large amount of fossil fuel is consumed to produce $H_2$ and it accounts for over 1% of global energy-related $CO_2$ emissions[3,4]. To solve the energy and environment-related problems in ammonia production, a method that operates at low temperatures and milder conditions is highly needed.

Electrochemical $N_2$ reduction reaction (NRR) produces ammonia cleanly and sustainably via the net reaction of $N_2 + (6H^+ + e^-) \rightarrow 2NH_3$ at ambient conditions. Various catalysts have demonstrated NRR activity at room temperatures, but the yield rate and the faradaic efficiency (<10%) are too low for economic production, due mainly to unwanted side reactions, i.e., hydrogen evolution reaction (HER)[5]. Theoretical studies have suggested that the theoretical limiting potential ($U_L$), where the all electrochemical elementary reaction steps become exothermic, for NRR, is about −1 V for various catalysts and is much more negative than $U_L$ of HER[6]. Thus, HER is expected to proceed before NRR when lowering the potential.

More quantitatively, however, potential-dependent measurements often showed that the NRR activity ($NH_3$ yield rate) begins to decrease even at a low overpotential region[7]. For numerous catalysts (Supplementary Note 1 and Supplementary Table S1), including transition metal[8–11], single-atom catalyst (SAC)[12–17], transition metal oxides[18,19], and non-metal catalysts[20–22], the maximum NRR activity (both faradaic efficiency and yield rate) has been generally observed with insignificant overpotentials, i.e., ~200 mV, then decreased at large overpotentials before reaching the mass-transfer limit. It results in the NRR current much smaller than the expected mass-transfer-limited values considering saturated $N_2$ concentration in aqueous solution at ambient conditions, i.e., ~1 mM[7].

The potential-dependent NRR activity is also different from other electrochemical reactions such as an electrochemical $CO_2$ reduction reaction ($CO_2RR$). $CO_2RR$ also competes with HER during the reaction. We compare the catalytic activity for NRR and $CO_2RR$ at Fe single-atomic site embedded at N-doped graphene (denoted as Fe@N₄), which was reported as an active catalyst for both NRR and $CO_2RR$[13,23,24]. Here we estimate $CO_2RR$ activity and NRR activity by using turnover frequency of CO formation and $NH_3$ yield rate, respectively. From our density functional theory (DFT) calculations, $U_L$ for $CO_2RR$ and NRR is −0.32 and −1.29 V, respectively (Fig. 1a). Here we calculated $U_L$ by the computational hydrogen electrode (CHE) model[25], which has been the most widely used method in estimating the energetics of electrochemical reactions. The $U_L$ is equal to the $\Delta G_{PDS}/e$, where $\Delta G_{PDS}$ is the free energy change at the most uphill individual step (i.e., potential-determining step (PDS)). The $CO_2RR$ activity increases with more negative potential and the maximum $CO_2RR$ activity is observed at around −0.7 V (Fig. 1b)[24]. At $U = -0.7$ V, which is more negative than $U_L$ (−0.32 V), $CO_2RR$ can be sufficiently facilitated and its activity begins to decrease due to approaching the mass-transfer limit. Thus, the potential-dependent $CO_2RR$ activity can be qualitatively explained by conventional DFT calculations. NRR activity also increases with more negative potential at first; however, it begins to decrease quickly at −0.4 V (pH = 7.2)[23] or −0.05 V (pH = 13)[13] (Fig. 1b), before reaching its $U_L$ (−1.29 V). This result indicates that NRR activity prematurely decreases with

increasing reduction potential, while its kinetics has not reached its expected theoretically maximum. NRR shows an unusual potential-dependent behavior that is unexplained by the conventional DFT calculations. Thus, the premature decrease in the NRR activity should be attributed to the intrinsic properties of catalysts.

The premature decrease of NRR activity indeed hampers to obtain reasonable $NH_3$ yield rate at the potential region where the NRR is expected to be sufficiently facilitated and should be the reason for the significantly lower $NH_3$ yield rates in all reported cases compared to other electrochemical reactions such as $CO_2RR$. In the case of Fe@N₄, e.g., the reported yield rate of CO formation and $NH_3$ formation is ~21 mmol h$^{-1}$ m$^{-2.24}$ and 0.562 mmol h$^{-1}$ m$^{-2.13}$, respectively. To understand such an unusual behavior of NRR, a fundamental understanding of potential-dependent changes in reaction energetics and coverage is required.

The possible reason for the premature decrease of NRR activity has been suggested qualitatively by the dominant H coverage at negative electrode potential, as the H binding increases faster than $N_2$ binding by the electrode potential[7,26,27]. Nørskov and colleagues[26] suggested that the surface will be covered by hydrogen rather than $N_2$ at negative enough potentials by using reaction equations. However, such a possibility has not been theoretically verified. To understand the premature decrease of NRR activity, a comprehensive understanding of the potential-dependent competition of NRR vs. HER should be investigated based on the quantitative change of the coverages and kinetics.

Here we attempt to quantify the effect of potential-dependent surface coverage on the NRR activity and unveil the origin of premature decrease of NRR activity. We note that although many theoretical studies investigated the details of NRR mechanisms on catalysts such as Ru[28,29], Fe[30], transition metal nitrides[31–33], and late transition metal surfaces[34], these studies did not investigate the potential-dependent behavior of NRR discussed above and used neutral-state DFT calculations with the CHE model[25]. In the CHE model, due to the constant charge constraint, the work function (or chemical potential) of the system changes from reactants to transition states (TSs) (or final states (FSs)) and fractional charge transfer is not allowed. This makes the CHE model, albeit widely used and proven successful for designing new catalysts and enhancing our understanding, not suitable for interpreting the experimentally observed potential-dependent behaviors of electrochemical catalysis.

In this work, we use the constant electrode potential (CEP) model, which treats the electrode–electrolyte interface as a polarizable continuum with implicit solvation model[35,36]. In this model, the number of electrons is adjusted to guarantee different states to have the same work function in the grand canonical states. This method has been used to understand many electrochemical reactions[37–47]. We compute energetics of NRR and HER as a function of electrode potential ($U$) for a single Fe atom catalyst embedded in N-doped graphene (Fe@N₄) as a model system (but the generalized discussion for other catalysts are given later in the paper). The calculated potential-dependent reaction energetics are then used in the microkinetic modeling (MKM), to obtain the active site coverages and yield rate measurements to compare them with experiments. Remarkably, we find a potential-dependent crossover between the H binding and $N_2$-binding energies, leading to a crossover in the active site coverages and $NH_3$ yield rate behaviors, all consistent with experiments. Further analysis demonstrates that the latter crossover originates from the larger charge transfer in the *H formation than that of *N₂ and *NNH formation. Further calculations on other catalysts reveal that the larger charge transfer in *H compared to *N₂ and *NNH formation is indeed a

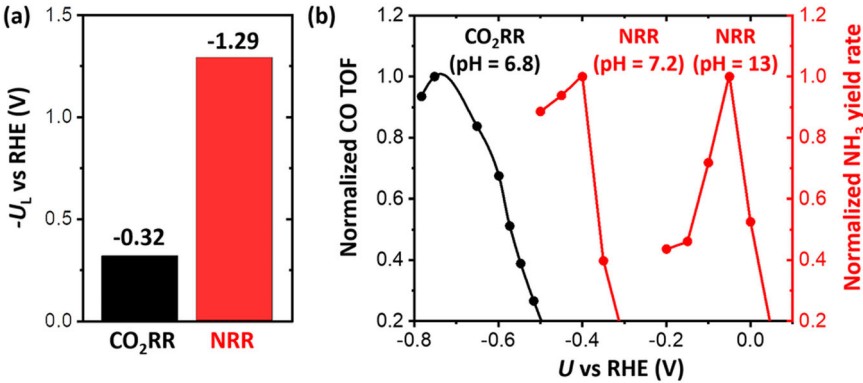

**Fig. 1 Comparison of onset potential for $CO_2RR$ and that for NRR on Fe@N$_4$ catalysts. a** $U_L$ for $CO_2RR$ (black, CO formation) and NRR (red, $NH_3$ formation) obtained on the Fe single-atom-embedded N-doped graphene using DFT calculations with the CHE model. **b** Potential-dependent measurements of turnover frequency (TOF) of CO in $CO_2RR$ (black) and $NH_3$ yield rate in NRR (red) taken from the literature; CO TOF (pH = 6.8) is taken from Ju et al.[24] and $NH_3$ yield rates at pH = 7.2 and 13 are taken from Lü et al.[23] and Zhang et al.[13], respectively. Normalized CO TOF and $NH_3$ yield rates are obtained by dividing their absolute values with its maximum.

general trend. We then discuss several directions to overcome this fundamental challenge of activity drop for NRR as a function of potential.

## Results

**Calculation models.** We choose Fe single atomic site anchored by four N atoms in the graphene (denoted as Fe@N$_4$) as a model system, as there are several well-characterized (including the isotope $^{15}N_2$ measurements) experimental results to compare[13,14,23]. As described above, Fe@N$_4$ catalysts showed volcano-shaped NRR activity with respect to the $U$. In the NRR measurements on Fe@N$_4$, the maximum $NH_3$ yield rate is obtained at $U = 0$, $-0.05$ V vs. reversible hydrogen electrode (RHE) in 0.1 M KOH[13,14] and $U = -0.40$ V (vs. RHE) in 0.1 M phosphate-buffered saline (PBS)[23]. To calculate the potential-dependent activation energy in electrochemical reactions, we include a hexagonal ice bilayer (H-down geometry)[48–50] above the Fe@N$_4$ site (Fig. 2). Here, three different reaction conditions (acidic, neutral, and alkaline) were considered. We use $H_2O$ as a proton donor in neutral and alkaline conditions, whereas we use solvated hydronium ion ($H_3O^+$) as a proton donor in acidic condition. Here we mainly discuss NRR under alkaline (pH = 13) and neutral conditions (pH = 7.2), the same conditions with the reported experiments[13,14,23]. However, we note that NRR under acidic conditions follows the same trend.

**Potential-dependent energetics.** We constructed the potential-dependent free energy diagram for NRR at 0, $-0.23$, and $-0.5$ V (vs. RHE at pH = 13) including activation energies by using the CEP model (Fig. 3). Calculation details for obtaining reaction energy and activation free energy under constant potential are shown in the "Methods" section and Supplementary Note 2. All possible reaction intermediates are listed in Fig. 2c. We note that highly exothermic reactions such as $*NH + H_2O \rightarrow *NH_2 + OH^-$ and $*NH_2 + H_2O \rightarrow *NH_3 + OH^-$ proceed barrierlessly. The lowest energy pathway based on the apparent activation energy is represented by a red line in Fig. 3. Here, the apparent activation energy is defined as the energy difference between the highest TS energy and the lowest energy intermediate in the catalytic cycle based on the energetic span model[51]. At 0 V, the $*NHNH_2$ formation shows the highest apparent energy (Fig. 3a), whereas $*NNH$ shows the highest apparent energy under negative $U$ (Fig. 3b, c). We found that the $*NNH$ formation begins to show the highest apparent activation energy at $U = -0.12$ V. Also, all proton-coupled electron transfer (PCET) reactions, except for the

$*NNH$ formation, becomes exothermic at $U = -0.5$ V. This result indicates that the $*NNH$ formation is a rate-determining step (RDS) under negative potential and a PDS. Thus, we will estimate the overall rate of NRR by using the $*NNH$ formation.

To investigate the effect of potential on $N_2$ and H coverages, we first compare the free energy change for $*N_2$ and $*H$ formation (* denoting the adsorbed species) on Fe@N$_4$ at different electrode potentials ($U$) (Fig. 4), the first reaction step for NRR and HER, respectively. $G_a(*H)$, $\Delta G(*H)$, and $\Delta G(*N_2)$ represent activation free energy for $*H$ formation, reaction free energy for $*H$ formation, and $N_2$ adsorption, respectively. The potential-dependent free energy diagrams for HER including activation energy are shown in Supplementary Fig. 4.

At 0 V, which is close to the equilibrium potential of NRR at standard state (0.057 V vs. RHE), $\Delta G(*N_2)$ is more negative than $\Delta G(*H)$ (Fig. 4a). With more negative potential, $\Delta G(*N_2)$, $G_a(*H)$, and $\Delta G(*H)$ all become more negative (favorable for reaction), but the $G_a(*H)$ and $\Delta G(*H)$ changes faster than $\Delta G(*N_2)$. Interestingly, $\Delta G(*N_2)$ changes by $U$, contrary to the general expectation that $N_2$ adsorption is a non-electrochemical reaction. This result indicates that $N_2$ adsorption is accompanied by partial electron transfer. The physical origin for electron transfer during $N_2$ adsorption is the $\pi$ back-bonding, which is the most important mechanism for $N_2$ binding at transition metal atom[52,53]. The back-donation of metal $d$ electrons to the lowest unoccupied molecular orbital of $N_2$ (antibonding $\pi^*$) weakens the N–N triple bond and activates $N_2$. Thus, the amount of charge transfer from metal atom to $*N_2$ is an important descriptor for estimating the extent of $N_2$ activation[54]. We found the increasing Bader charge density[55] and the elongated N–N bond length in $*N_2$ with more negative $U$, suggesting that more negative $U$ promotes $N_2$ activation via increasing back-donation (Supplementary Fig. 5).

The slope ($\Delta G$ vs. $U$) increases in the order of $\Delta G(*N_2)$ (0.14) < $G_a(*H)$ (0.76 and 0.96) < $\Delta G(*H)$ (1.14) (Fig. 4a, b). The physical meaning of the slope, how sensitively $\Delta G$ (or $G_a$) changes with $U$, is the amount of electron transfer during reaction and we will discuss it in more detail later. Due to the difference in slope, a crossover in which $\Delta G(*N_2) = \Delta G(*H)$ occurs at potential $U_{cross} = -0.15$ V. This result would indicate a strong dependency of $N_2$ coverage ($\theta_{N_2}$) on $U$. At $U > U_{cross}$, $\theta_{N_2}$ will be higher than H coverage ($\theta_H$), whereas at $U < U_{cross}$, H coverage could overwhelm $\theta_{N_2}$ and hinder the NRR. Actual coverages as a function of potential are calculated and discussed in further detail with MKM in the next section.

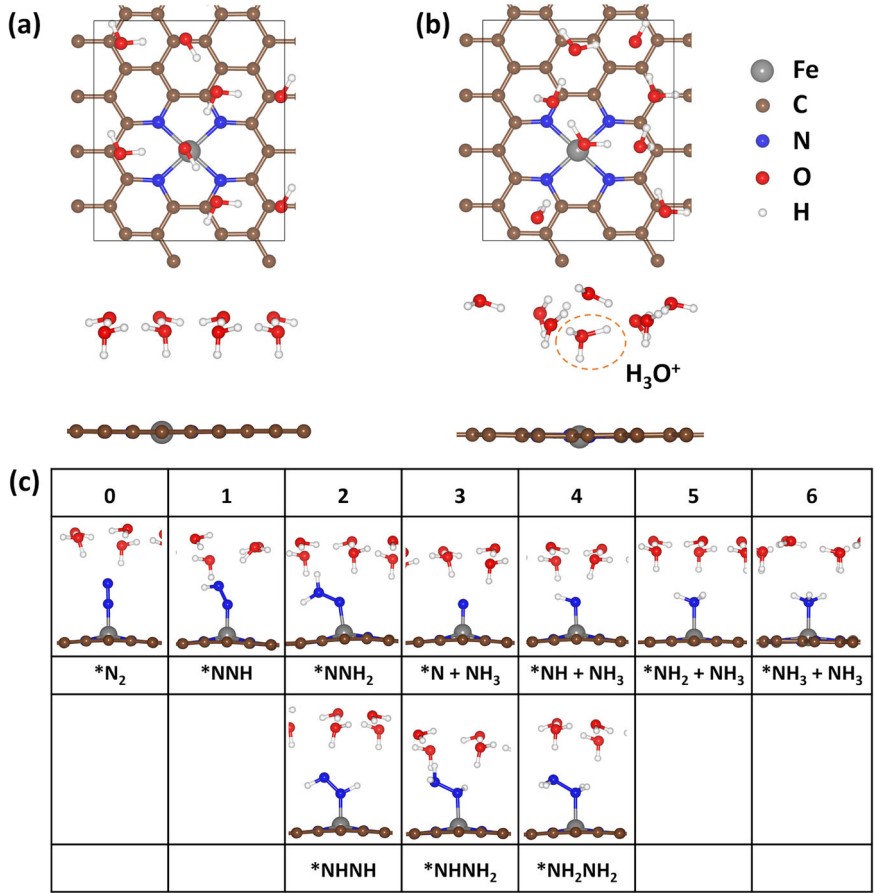

**Fig. 2 Calculation models for Fe@N$_4$ catalysts.** Fe@N$_4$ with **a** a hexagonal ice bilayer water and **b** a hexagonal ice bilayer water containing a solvated H$_3$O$^+$. Top view and side view are shown in the upper panel and lower panel, respectively. **c** The optimized geometries of all possible reaction intermediates of NRR. The number of transferred protons is listed in the first row. For *NH + NH$_3$, *NH$_2$ + NH$_3$, and *NH$_3$ + NH$_3$, the liberating NH$_3$ is omitted for the clarity.

Next, we analyze the trend of NRR and HER activity by comparing $G_a$(*N$_2$ → *NNH) and $G_a$(*H), an RDS under negative $U$ and PDS in NRR and HER on Fe@N$_4$ (Fig. 3 and Supplementary Fig. 4), respectively. Here, $G_a$(*N$_2$ → *NNH) (or $\Delta G$(*N$_2$ → *NNH)) represents activation energy (or reaction energy) of *N$_2$ + (H$^+$ + $e^-$) → *NNH. The potential-dependent free energy diagrams for *NNH formation are listed in Supplementary Fig. 6. The reaction pathways for the *H formation and *NNH formation are shown in Fig. 5, which proceed via H$^+$ transfer from water and reorganization of water layer and adsorbate. We find that the TS structure of the *NNH formation is especially similar to its FS, as the *NNH formation is highly endothermic (Figs. 4 and 5). This agrees with the Hammond postulate[56], i.e., TS of endothermic reaction resembles FS (and vice versa).

We find that $G_a$(*N$_2$ → *NNH) (or $\Delta G$(*N$_2$ → *NNH)) decreases with negative $U$, indicating that the NRR activity would increase with more negative potential (Fig. 4a, b). However, $G_a$ and $\Delta G$ for *NNH formation are higher than those of *H formation. For both proton donors (H$_3$O$^+$ and H$_2$O), $G_a$(*N$_2$ → *NNH) is higher than $G_a$(*H) (Fig. 4b) and their differences get even larger with more negative potential (Supplementary Note 3 and Supplementary Figs. 9 and 10). This result indicates that the rate of NRR is lower than that of HER in both acidic and alkaline conditions, and the rate of HER with potential grows even faster than that of the NRR rate with potential. Consequently, the hindering effect of HER would become increasingly more important with a negative potential.

Interestingly, the slope for $\Delta G$(*N$_2$ → *NNH) vs. $U$ at low overpotential region (from 0 to −0.5 V vs. standard hydrogen electrode (SHE)) is different from that at higher overpotential region (from −0.5 to −1 V vs. SHE) (Fig. 4a and Supplementary Fig. 6c). This result arises from the significant change in the *NNH geometry with $U$ (Supplementary Fig. 7a). We optimized all structures with the proper number of electrons in the slab model and found that *NNH optimizes to a more bent structure with negative potential. When the geometry of * and *NNH is fixed at their optimized geometries at neutral state, the slope ($\Delta G$(*NNH) vs. $U$) is constant (Supplementary Fig. 7b). This result indicates that the geometry relaxation by $U$ leads to the potential-dependent charge transfer. All energetics and associated charge transfer in Fig. 4a, b are listed in Supplementary Tables 3 and 4.

In the CHE model, the slope for N$_2$ adsorption and PCET step are constant at 0 and 1, respectively. However, the slope obtained by the CEP model is quite different from that of the CHE model (Fig. 4). We find that reaction energy obtained by the CEP model is also different from that obtained by the CHE model at the same potential (Supplementary Fig. 11). Such a deviation of slope or adsorption energy in the CEP model have been reported[39,43–45]. The main physical origin for such a deviation from the CHE model, we suggest, is the change in the potential of zero charge ($U_{PZC}$) during chemical reaction. The $U_{PZC}$ is obtained by the electrode potential of the slab model at neutral state. At neutral state, the $U_{PZC}$ (or work function) changes during chemical reaction[57]. Thus, extra (or deficient) electrons are involved in the

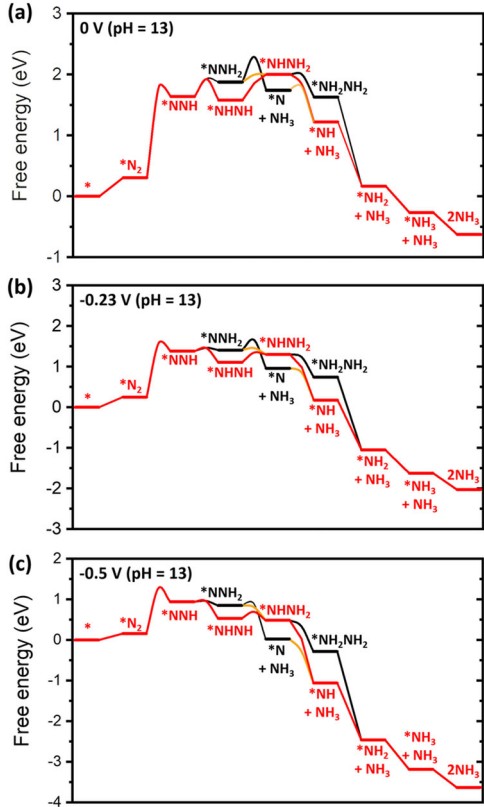

**Fig. 3 Free energy diagram of NRR including activation energy.** Free energy diagram of NRR (pH = 13) at **a** $U = 0$ V vs. RHE, **b** $U = -0.23$ V vs. RHE, and **c** $U = -0.5$ V vs. RHE. The lowest activation energy requiring reaction pathway is represented by red line.

CEP model to compensate the change of $U_{PZC}$ during reaction. For example, the $U_{PZC}$ of Fe@N₄ (denoted as $U_{PZC}(*)$) is −0.83 V, and that of *H ($U_{PZC}(*H)$) is −0.54 V (Supplementary Fig. 12a). To set the $U$ of * and *H to 0 V, electrons amounting to 0.83 V is extracted in *, whereas electrons amounting to 0.54 V is extracted in *H. Consequently, extra electrons corresponding to 0.29 V (0.83–0.54) are engaged to compensate the change of $U_{PZC}$. In the CHE model, all reaction intermediates at neutral state are assumed to be at the same $U$ and, thus, change of $U_{PZC}$ during reaction is not considered. We find that the change of $U_{PZC}$ (denoted as $\Delta U_{PZC}$) during reaction well correlates with the deviation from the CHE model (Supplementary Figs. 12 and 13), similar to the previous work[43]. The detailed discussion for the relation between $U_{PZC}$ and the deviation is in Supplementary Note 4.

**Microkinetic modeling.** To further investigate how the surface coverages and NRR activity change by $U$, we performed an MKM based on the potential-dependent energetics of NRR and HER obtained from the CEP model (Fig. 6a) described above. For HER, only the Volmer–Heyrovsky reaction is considered, as the Volmer–Tafel pathway is much less active on Fe@N₄ (Supplementary Note 5 and Supplementary Figs. 14 and 15). The details on the MKM and energetics are in Supplementary Note 6. As the NRR activity was measured at pH = 13 (0.1 M KOH)[13,14] and pH = 7.2 (0.1 M PBS)[23] in the experiments (Fig. 1), we note that the MKM results under alkaline, neutral, and acidic conditions are represented in RHE scale at pH = 13, 7.2, and 0, respectively. The pH in our simulation is assessed by considering the change in

the activity of ions (H⁺ or OH⁻) in bulk electrolyte (i.e., bulk pH), which are the reference state energy of the PCET step.

We note that the bulk pH and pH near the active site (i.e., local pH) are different due to the accumulated ions (e.g., OH⁻) at the interface during the reaction. To fully investigate the effect of local pH on energetics, pH should be explicitly considered in the DFT calculations, yet highly challenging due to the computational cost in large-scale explicit simulations. Instead, previous studies assessed the pH effect by considering the change in the activity of ions (the method used in our study) and were able to reproduce the experimental trend[58–60]. Here we use relative NH₃ yield rate rather than absolute one due to the inherent uncertainties in quantifying reaction rates from both experiment and theory[39]. The relative NH₃ yield rate is obtained by normalizing the values by its maximum.

For all reaction conditions, we find that the relative NH₃ yield rate (denoted as $r_{NH_3}$) has a volcano shape at the low overpotential region (Fig. 6a), similar to the experiments (Fig. 1). The maximum $r_{NH_3}$ is obtained at −0.275 (pH = 13), −0.575 V (pH = 7.2), and −0.20 V (pH = 0). Interestingly, this $U$ at maximum $r_{NH_3}$ (−0.20 ~ −0.575 V) is highly more positive than theoretical limiting potential ($U_L$) for NRR of −1.29 V (Supplementary Fig. 11), indicating that the $r_{NH_3}$ decreases prematurely even before reaching the theoretical limiting potential needed to drive the reaction. Noticeably, the $\theta_{N_2}$ also has a volcano shape and the $U$ at maximum $\theta_{N_2}$ is very similar, differing by ~0.1 V from the $U$ at maximum $r_{NH_3}$ (Supplementary Table 5). This result indicates that the premature maximum in $r_{NH_3}$ at substantially more positive potential originates from the decrease in $\theta_{N_2}$. Contrary to $\theta_{N_2}$, $\theta_H$ increases continuously and we find a crossover in coverages between $\theta_{N_2}$ and $\theta_H$ as predicted by the crossover between $\Delta G(*H)$ and $\Delta G(*N_2)$. Due to the fast kinetics of the Heyrovsky reaction (Supplementary Fig. 4), *H would be easily eliminated and the crossover in coverages between $\theta_{N_2}$ and $\theta_H$ is not observed at pH = 0 (Fig. 6a). However, we emphasize that the premature decreases of $r_{NH_3}$ and $\theta_{N_2}$ are consistently observed at alkaline, neutral and acidic conditions, indicating that such a phenomenon occurs pH-independently.

To further estimate the effect of HER on potential-dependent behavior of NRR, we performed the MKM simulations without considering HER, corresponding to an ideal environment in which NRR proceeds without competing with HER. The $r_{NH_3}$ without HER is obtained by normalizing the values with the maximum value of $r_{NH_3}$ with HER. Without the HER, the premature decrease of $r_{NH_3}$ and $\theta_{N_2}$ are not observed (Fig. 6a). We find that the $r_{NH_3}$ and $\theta_{N_2}$ continuously increases with negative $U$ in all reaction conditions, clearly suggesting that the occurrence of early maximum in $r_{NH_3}$ originates from the decreasing $\theta_{N_2}$ by the competing HER. Furthermore, $r_{NH_3}$ and $\theta_{N_2}$ obtained by MKM without HER are higher than those with HER at all potential ranges, confirming that the HER indeed hampers the NRR.

We are now in a position to understand the origin of premature decrease of $r_{NH_3}$ and potential-dependent competition with HER. At lower overpotentials where the $\theta_{N_2}$ is more dominant than $\theta_H$, increasing $\theta_{N_2}$ and decreasing $G_a(*N_2 \rightarrow *NNH)$ with negative $U$ results in the increasing $r_{NH_3}$ (Figs. 4b and 6a). At higher overpotentials, although $G_a(*N_2 \rightarrow *NNH)$ continuously decreases, the $\theta_H$ becomes high enough to block active sites and reduces $\theta_{N_2}$. Consequently, the electrode potential at maximum $r_{NH_3}$ is close to the potential at maximum $\theta_{N_2}$ (Supplementary Table 5), and after reaching the maximum, the $r_{NH_3}$ decreases due to decreasing $\theta_{N_2}$ with more negative

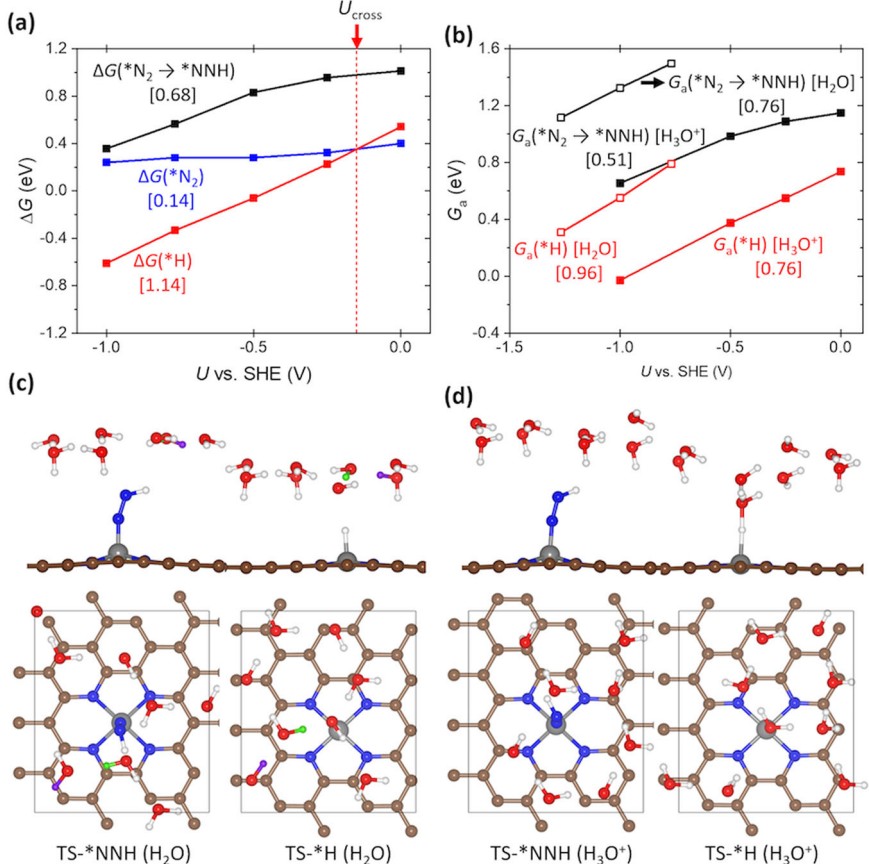

**Fig. 4 Potential-dependent energetics of *H, *N$_2$, and *NNH formation. a** Change of $\Delta G(*N_2 \rightarrow *NNH)$, $\Delta G(*N_2)$, and $\Delta G(*H)$ by $U$. The red vertical dashed line in **a** represents the crossover potential ($U$ at $\Delta G(*H) = \Delta G(*N_2)$). **b** Change of $G_a(*N_2 \rightarrow *NNH)$ and $G_a(*H)$ by $U$. The slope ($\Delta G$ vs. $U$ or $G_a$ vs. $U$) of each reaction is shown in the graph. Open and closed squares represent $G_a$ obtained by using $H_2O$ and $H_3O^+$ as a proton donor, respectively. Black, blue, and red lines represent reaction energetics for *NNH, *N$_2$, and *H formation, respectively. The optimized transition state geometries of *NNH formation and *H formation **c** at 0 V vs. RHE (pH = 13) by using $H_2O$ and **d** those at −0.5 V vs. RHE (pH = 0) by using $H_3O^+$.

potential. These results clearly demonstrate that potential-dependent $\theta_{N_2}$ is the underlying mechanism for the potential-dependent $NH_3$ yield rate behavior.

In the MKM simulations, the maximum $r_{NH_3}$ is observed at $U = -0.275$ V (pH = 13) and −0.575 V (pH = 7.2), respectively. This is similar to the experimental value at pH = 13 (0.0 and −0.05 V)[13,14] and pH = 7.2 (−0.40 V)[23], qualitatively explaining the experimental trend. Due to the intrinsic DFT error especially significant in ionic species[61] and different environments from experiments, such as explicit electrolytes and local pH, we note that such a difference is acceptable. However, a sharp increase and decrease of $r_{NH_3}$ at low overpotential region are well reproduced in our MKM simulations. Interestingly, $U$ at maximum $r_{NH_3}$ obtained by the MKM is significantly more positive than the $U_L$ for NRR (−1.29 V). This suggests that considering the $U_L$ only is insufficient to fully understand NRR behavior and explains why the reported theoretical $U_L$ has been disagreed with experiments[27], but instead, the potential-dependent competition between NRR and HER should be considered.

To verify the critical effects of potential-dependent binding energies, we also performed the MKM simulations using the binding energies obtained by the CHE model (Fig. 6b and Supplementary Note 7). Here we used 0.5 as a charge transfer in all TS, which is a reasonable assumption in the Heyrovsky-like reaction[30,34,47,62]. We find that the MKM simulations using the

CHE model do not agree with the experiment. The premature decrease of $r_{NH_3}$ and the crossover of active site coverages are not observed. In the CHE model, although the $\Delta G(*H)$ also becomes more negative with $U$ as in the CEP model, the potential dependence of the reaction energetics is assumed to be the same as long as the reaction involves the same number of electrons. That is, both the Volmer (*H formation) and Heyrovsky (*H elimination) reactions formally involving $1 - e$ transfer have the same potential-dependent behaviors. As a result, the accumulation of the H coverage is not observed with potential. To further verify it, we plotted the MKM results using the CHE model for a hypothetical case (inspired by the CEP results) in which the activation energy of Volmer and that of Heyrovsky reactions have different slopes of 0.9 and 0.5, respectively, and compared the results with the case of constant slopes (Supplementary Fig. 17). Indeed, the H-coverage crossover with potential is only observed in a hypothetical case in which the slopes for Volmer and Heyrovsky reactions are different, a situation that only the CEP model can treat rigorously (Fig. 4). These results indicate that the potential dependence of all electrochemical and non-electrochemical reactions should be rigorously calculated to describe properly the electrochemical catalytic activity of NRR competing with HER.

**Origin and descriptor for different slopes.** In the previous sections, we found that the change of $\Delta G(*H)$ with potential ($\Delta G$

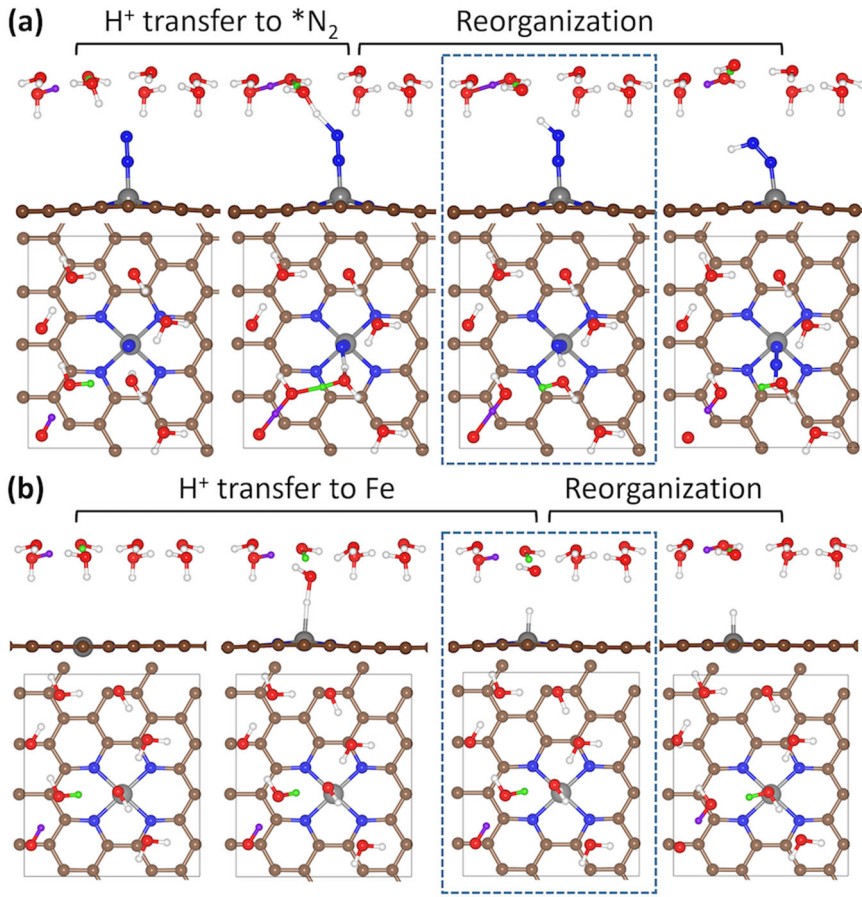

**Fig. 5 Reaction pathways of *H formation and *NNH formation.** Reaction pathway for **a** $^*N_2 + H_2O \rightarrow {}^*NNH + OH^-$ and **b** $^* + H_2O \rightarrow {}^*H + OH^-$. Side view and top view are listed in the upper panel and lower panel, respectively. Transition state of each reaction is highlighted with blue dashed box. Green and purple balls represent the transferred H atoms during reorganization.

vs. $U$) with a steeper slope than $\Delta G(^*N_2)$ and $\Delta G(^*N_2 \rightarrow NNH)$ leads to the surface coverage crossover and premature decrease of NRR activity. To understand the origin of these different slopes, we consider the fundamental Nernst equation. From the Nernst equation [$\Delta G$ (in eV) $= -\Delta N_e U$], the first derivative of the $\Delta G$ with respect to $U$ is $d(\Delta G)/dU = -\Delta N_e$, where the $\Delta N_e$ represents the amount of transferred electrons in the reaction, and thus we compared the average value of $\Delta N_e$ (denoted as $\Delta \bar{N}_e$) with the computed slopes obtained for key reactions ($\Delta G(^*H)$, $\Delta G(^*N_2)$, and $\Delta G(^*N_2 \rightarrow {}^*NNH)$). Details on calculating slope and $\Delta \bar{N}_e$ are in Supplementary Note 8. We further consider other catalysts such as Ru(0001), Rh(111), Fe(110), Ru@N₃, Ru@NC₂, Fe@N₃, and Ag@N₄. The latter catalysts are chosen, as their NRR activity and their volcano-like trend were experimentally observed (Ru nanoparticles[8], Rh nanosheet[11], Fe foil[10], and M-NC catalysts[15–17,23]). As expected, the computed $\Delta \bar{N}_e$ is in excellent agreement with the slope for all catalysts considered here, indicating that $\Delta \bar{N}_e$ determines the potential-dependent adsorption behavior (Supplementary Figs. 18 and 19, and Supplementary Table 6). This result indicates that the reactions with more electron transfer become energetically more favorable as the potential becomes more negative. Thus, we compare the amount of electron transfer ($\Delta N_e$) at 0 V (vs. SHE) as a representative to estimate the slope (Fig. 7).

We find that the $\Delta N_e$ is highly deviated from that of the CHE model (0 for N₂ adsorption and 1 for PCET step) in several catalysts (Fig. 7), which mainly originates from the change of $U_{PZC}$ during reaction. A linear relationship between $\Delta U_{PZC}$ with the deviation in $\Delta N_e$ is obseved for various catalysts

(Supplementary Fig. 20). This result indicates that catalysts, whose $U_{PZC}$ easily changes during a chemical reaction, require extra (or deficient) electrons during electrochemical reactions. We also note that a large deviation in the slope (or charge transfer) has also been reported on SAC[39,44] and N-doped graphene[45], which are incorporated in our system (Fe@N₄).

Interestingly, the $\Delta N_e$ increases in the order of $\Delta G(^*N_2) < \Delta G(^*N_2 \rightarrow {}^*NNH) < \Delta G(^*H)$ for all catalysts, indicating that the key reactions for NRR (N₂ adsorption and *NNH formation) involve fewer electrons than *H formation (Fig. 7). These catalysts will show the coverage crossover as discussed above, as the tendency of *H formation increases faster than that of *N₂ and with more reduction potential, resulting in the early drop of NRR activity. Thus, the coverage crossover is an intrinsic property of active site and generalizes in various catalysts. Here we estimate the overall kinetics of NRR by using *NNH formation, the PDS of Fe@N₄, as well as various catalysts. However, catalysts with strongly N-binding affinity, which lies on the left leg of the NRR volcano plot, are limited by $^*NH_2 + (H^+ + e^-) \rightarrow NH_3$[6,63]. Thus, we note that the potential-dependent energetics and charge transfer associated with *NH₂ should be considered for strong N-binding catalysts.

The control of reaction selectivity between NRR and HER has been extensively studied in fields of biochemistry, bio-electrochemistry, molecular catalysis, and electrochemistry[64–66]. It has also been known that natural N₂-fixation catalyst, e.g., FeMo-cofactor, performs the NRR with a significant reaction selectivity up to 75%, in subtly controlled organisms[67,68]. In electrocatalysis, the design of the entire catalytic system, including catalytic active surfaces, supporting promoters, electrolytes, and reaction

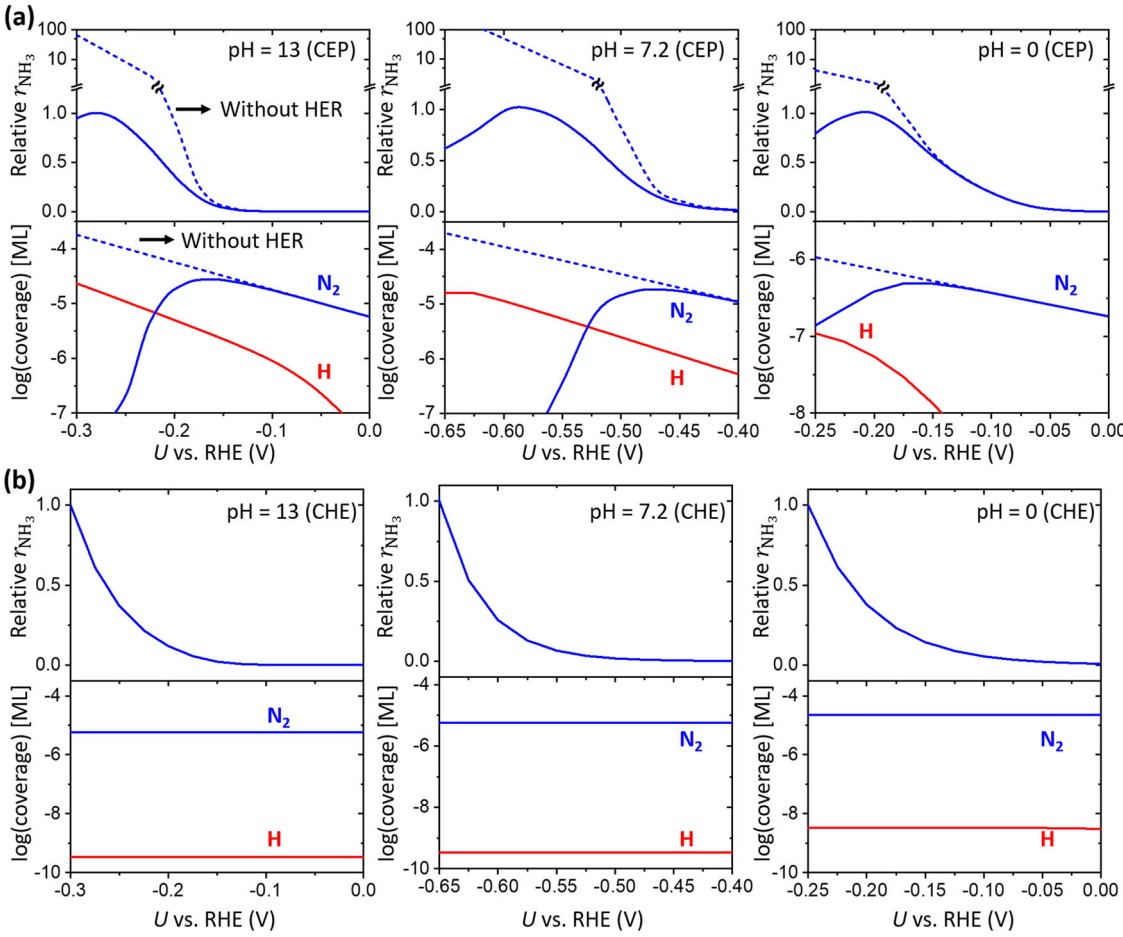

**Fig. 6 The change in $\theta_{N_2}$, $\theta_H$, and $r_{NH_3}$ by $U$ obtained by the MKM. a** MKM results using the CEP model and **b** MKM results using the CHE model at three different pH (pH = 13, 7.2, and 0). The relative $r_{NH_3}$ is obtained by dividing the $r_{NH_3}$ by its maximum. Dashed lines represent MKM results without HER. The relative $r_{NH_3}$ and coverage are shown in upper and lower panels, respectively.

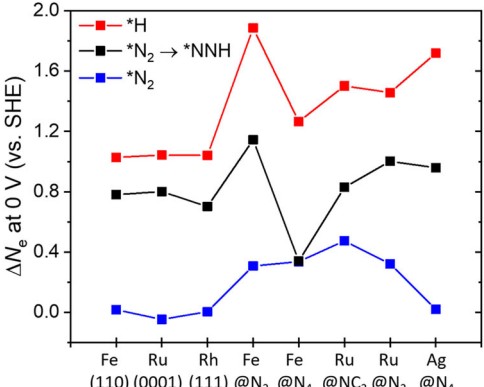

**Fig. 7 The amount of electron transfer ($\Delta N_e$) in *H, *N$_2$, and *NNH formation at 0 V.** Black, red, and blue colors represent *NNH formation from *N$_2$, *H formation, and *N$_2$ formation, respectively.

conditions, must also be tuned to achieve a considerable NRR selectivity and to overcome the coverage crossover between *N$_2$ and *H. As the fine-tuning of NRR selectivity over HER has been demonstrated in metal-complex catalysis, more delicate design of binary, tertiary, or multi-component electrocatalysts can regulate the *H formation over the *N$_2$ or *NNH production[65]. More practically, controlling the concentration of N$_2$ and H$^+$ at the electrode–electrolyte interface with increasing the *H formation

barrier could be helpful, such as by the coatings with a hydrophobic layer[69], utilizing polar aprotic ionic solvent (high N$_2$ solubility)[70,71], using gas diffusion electrode (high N$_2$ concentration)[72], or using bulky proton donor in non-aqueous electrolytes[26,73].

## Discussion
In this study, we investigated the origin of decreasing NRR activity with potential, a major obstacle to practical NRR, generally occurring in most heterogeneous metal catalysts. The key aspect is shown to be the potential-dependent crossover in the H- vs. N$_2$-binding affinities and associated active site coverages (initially favoring *N$_2$ but, with more negative potential, favoring *H). The degree of charge transfer involved in the respective reaction, consistent with the Nernst equation, is responsible for the crossover behavior in general for various catalysts. We expect the degree of charge transfer to be a simple and general descriptor to understand other electrochemical reactions and their potential dependency, such as CO$_2$ reduction to various products[74] and oxygen reduction reaction to H$_2$O vs. H$_2$O$_2$[75].

## Methods
**Computational details.** All calculations were performed using spin-polarized DFT methods implemented in the Vienna Ab initio Simulation Package (VASP) with projector-augmented wave pseudopotential[76–78]. We used the revised Perdew-Burke-Ernzerhof (RPBE) functional developed by Hammer et al.[79,80]. Cutoff energy was set to 400 eV. The convergence criteria for the electronic energy difference and forces are $10^{-5}$ eV and 0.05 eV/Å, respectively. TS optimization was

performed using the climbing image nudged elastic band (CI-NEB)[81,82] combined with the improved dimer method (IDM)[83,84]. We first obtained the initial TS geometry by the CI-NEB and verifying a first-order saddle point by performing vibrational analysis. Then, we performed IDM by varying the number of electrons to tune the potential. The effect of pH on energetics is incorporated by changing the chemical potential of $H^+$ or $OH^-$ in bulk. Details in calculating the potential-dependent activation energy are in Supplementary Note 2.

**Constant electrode potential model**. The electrode potential ($U$) referenced to that of SHE is given by

$$U = \frac{-\mu(e^-) - \Phi_{SHE}}{e} \qquad (1)$$

where $\mu(e^-)$ and $\Phi_{SHE}$ represent the chemical potential of electron and work function of the SHE, respectively. We used 4.43 eV for $\Phi_{SHE}$, obtained by the RPBE[85]. Also, the 4.43 eV lies within the experimentally obtained $\Phi_{SHE}$ (4.44 ± 0.02 eV)[86]. The $\mu(e^-)$ is equal to the Fermi level compared to the electrostatic potential at bulk electrolyte. In the CEP model, the number of electrons ($N_e$) is adjusted to guarantee different states to have the same $\mu(e^-)$ in the grand canonical states. Thus, we adjust $N_e$ to match the $U$ of the slab model with target potential ($U_{target}$). We set the convergence criteria for $U$ as $|U - U_{target}| < 10^{-4}$ V.

If the two different slab model have identical $\mu(e^-)$, while they have different $N_e$, grand canonical electronic energy ($\Omega$) is given by[46,47]

$$\Omega = E_{DFT} - N_e\mu(e^-) \qquad (2)$$

$E_{DFT}$ represents the electronic total energy of the slab model (in non-zero charge state). We obtained free energy ($G$) by using thermal energy correction to $\Omega$.

We treat the electrode–electrolyte interface as a polarizable continuum via the linearized Poisson–Boltzmann equation, which is implemented in VASPsol[35,36], an extension of VASP. In this method, ionic counter-charges are implicitly placed at the interface. The net dipole originating from adding (or extracting) electrons is screened by the ionic counter-charges. We set the Debye length by 3 Å, corresponding to a 1 M concentration of electrolyte. The relative permittivity of the bulk solvent was chosen as that of water (78.4). We further added $QV$ correction to the electronic energy, where the $Q$ and $V$ represent the net charge of the slab model and the negative value of the electrostatic potential at bulk electrolyte, respectively. This correction is a missing contribution to the total energy in the present VASPsol release and makes that the $\Delta G$ become independent on cell size.

**Calculation model**. The M@$N_xC_y$ site is constructed by using a rectangular graphene supercell containing 32 carbon atoms. The transition metal surfaces are modeled by ($3 \times 3$) atomic supercell with four layers. The bottom two layers were fixed to their optimized bulk positions, whereas other atoms were fully relaxed. All slab models include more than 18 Å of vacuum in the $c$-axis. The ($4 \times 3 \times 1$) and ($3 \times 3 \times 1$) Monkhorst-Pack mesh of $k$-points[87] were sampled for M@$N_xC_y$ and transition metal surfaces, respectively.

## Data availability
The main data supporting the findings of this study are contained within the paper and its associated Supporting Information. All other data are available from the corresponding author upon request.

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

## Acknowledgements

We acknowledge generous financial support from NRF Korea (NRF-2016M3D1A1021147, NRF-2019M3D1A1079303) and supercomputing time from Korea Institute of Science and Technology Information (KISTI).

## Author contributions

C.C. and Y.J. conceptualized the idea and wrote the manuscript. C.C. performed all DFT calculations and analyzed the results. G.G. performed MKM simulations. J.N. and H.S.P. assisted with the interpretations of the results. All authors discussed the results and assisted during manuscript preparation.

## Competing interests

The authors declare no competing interests.
