## [Peer Review File · Nature Communications]

REVIEWER COMMENTS

Reviewer #1 (Remarks to the Author):

In the manuscript titled "Understanding Potential-dependent Competition Between Electrocatalytic Dinitrogen and Proton Reduction Reactions", Choi et al. reported a computational study of nitrogen reduction reaction (NRR). By combining density functional theory (DFT) calculations and kinetic model, the authors concluded that the reason for the early loss of NRR efficiency attributes to the loss of N₂ adsorption when H adsorption becomes stronger at more negative applied potential. Thus, hydrogen evolution reaction (HER) becomes dominant. Overall, it looks to me that this manuscript is worth publishing. But it seems that the evidence currently provided by the authors cannot fully support their conclusions as they expected. Based on my expertise and understanding of NRR and HER. I raise several comments for the authors to consider as follows.

1. I am more skeptical of the calculation results shown in Figure 5. According to the hypothesis proposed by the authors, competitive adsorption between N₂ and H* is the major cause of the competition between NRR and HER. More specifically, N₂ adsorption is less dependent on voltage. Therefore, the relationship between the adsorption energy and the voltage change is not obvious. Instead, the adsorption of H highly depends on voltage. Therefore, its adsorption energy has a significant correlation with voltage. The above analysis can explain the calculation results in Figure 4, but not Figure 5. According to the above assumptions, Figure 5 should also appear similar trend as shown in Figure 4. Although the author emphasizes that CHE has limitations in describing charge transfer for transition states, the CHE model should capture the above-mentioned dependence of adsorption energy and voltage. I can't understand the significant differences between Figure 4 and Figure 5. I suggest the authors add more discussions and explanations.

2. What is the physical meaning of dG/U in Figure 6? Is it a capacitance?

3. I cannot agree with the authors in the discussions of Figure 1. Both of the above reactions are multiple-step reactions (NRR is 6 e- and HER is 2e-). The overall reduction potential, in most cases, is known to be not related to the performance. Therefore, I can not be convinced that "NRR should have more negative potential than CO₂RR" as the authors concluded.

Reviewer #2 (Remarks to the Author):

Choi et al used electronic structure calculations in order to understand the competition between electrochemical ammonia production and hydrogen evolution on a single-site iron catalyst. This is an interesting and important research topic and I believe the use of grand-canonical DFT methods can provide significant new insights to processes like these. Unfortunately, I found this work to suffer from many severe flaws (discussed below) and to not provide meaningful insights to the literature.

Major flaws:

- The free energy diagrams found in Figure SI-3 form the basis for most of this manuscript. Here, they compare the diagrams produced from the well-known computational hydrogen electrode (CHE) model to what they are referring to as the constant-electrode potential (CEP) model. It is shocking how different the results are between the two models, especially since the CHE model is thermodynamically exact in the limit of integer electron transfer (or alternatively, of zero change in surface-normal dipole). If their CEP model is thermodynamically consistent, I would expect deviations from the CHE model to be small, perhaps on the order of 0.1 eV (for example, if an electron transfer was 0.9 instead of 1). Yet we can observe dramatic differences: for example, the step *NHNH → *NHNH₂ appears uphill by ~0.5 eV in the CHE model and downhill by about the same amount in the CEP model. This implies that CHE made a 1-electron error in this step! This is hard to believe; if this is really the case, the authors need to fully explain the difference and the physical meaning of such.

- The mechanism they have employed is not justified. There are many possible pathways that this

reaction could follow, even on a single-site catalyst, and the authors have only listed one, without justification.

- Related, the free energy diagrams have a step that appears to liberate NH_2 . ($\text{*NH}_2\text{NH}_2 \rightarrow \text{*NH}_2 + \text{NH}_2$) This is a free radical! This seems very unlikely. Why was only this single mechanism explored? There are many ways that N-N bond could be broken, and I do not expect the liberation of a radical from the surface would be the most favorable way. (Additionally, they calculate the liberation of this radical to be exergonic, which seems suspect.)

- In Figure 2, the transition states corresponding to part (c), (e) and (f) look quite similar to the final states. I.e., I do not see any significantly stretched or distorted bonds. How do the authors confirm that TS obtained are the true transition states? For clarity and better understanding, the authors should include the charge/electron transfer associated between the IS, TS and FS, as well as potential-energy diagrams of the minimum energy pathway.

- If I understand it correctly, Figure 3(a) is pure thermodynamics and should align with CHE in the limit of integer electron transfer; that is, deviations from integer slope should be caused by non-integer electron transfer. To what do the authors attribute a ~ 0.2 e transfer for *N_2 adsorption? I would have suspected that this elementary step doesn't involve significant charge transfer. Similarly, why are the charge transfers associated with the other two steps not closer to 1? [As a more minor point, how are the slopes calculated? The data points are not connected by a single straight line.]

- Similarly, Figure 6 does not make sense to me. (The authors are plotting nearly the same quantity in the x and y directions.) It is bizarre to see a huge window in the charge transfer associated with *N_2 adsorption (ranging from 0 to almost 0.6 e). How do the authors justify a charge transfer of 0.6 e for a step that doesn't involve explicit charge transfer? Similarly, for *H adsorption, the charge-transfer window varies over a huge window from roughly 1 to 2 e; how could the authors justify the 2-electron case (almost equivalent to the total electrons transferred in the entire HER)?

- The microkinetic results seem are suspect, since they didn't calculate barriers for many of their steps. (It seems they only have barriers for the couple of steps that they declared to be "potential determining" in the manuscript; I do not see others reported.) I personally would say that all barriers should be included in such a model since they are now rather straightforward with constant-potential methodologies; although I agree that it's likely that setting some to zero won't affect the kinetics. However, I can see a couple of apparently significant omissions just from looking at the free energy diagrams:

* At 0 V, the highest-energy intermediate is *NHNH_2 or *NHNH , depending on if you examine the CHE or CEP model. (As noted earlier, the difference in elementary thermodynamics between these two approaches is troubling.) Therefore, it's a good guess that the largest **overall** barrier (measured from the most-abundant reactive intermediate, MARI) would be from either the formation or destruction of these high-energy species. The largest **overall** barrier, not the largest individual barrier, is what typically best approximates the kinetics, so the omission of these barriers is significant.

* There are non-electrochemical steps involved, and notably $\text{*NH}_2\text{NH}_2 \rightarrow \text{*NH}_2 + \text{NH}_2$. Since this step will not be affected by potential (to a first order), its barrier will not diminish with potential as the researchers are implicitly assuming other steps will. Therefore, this barrier and ones like it must be calculated. It is a bit strange to examine N_2 reduction without examining breaking of the N-N bond.

- They seem to give two different explanations for the H_2 production overwhelming N_2 reduction (which is advertised to be the significant conclusion of this manuscript). On page 8, they attribute it to the higher coverage of H as the potential gets more negative (apparently blocking sites for N_2 adsorption). Later on the same page they seem to be making the argument that it is the barrier to form *NNH , as compared to the barrier for *H , that limits. Which is it? I don't expect two different steps to be equally responsible.

Significant issues:

There are many other significant issues with this manuscript, which are too numerous to mention. I will list some of them here, but I do not mean to imply this is a complete list.

- The free energy diagrams should be in the manuscript itself; it's too confusing to have to look to ill-explained SI figures to follow the logic of the discussion in the manuscript.

- In Supplementary figure 6, why is the red plot (-0.5 V) not half-way between the black (0 V) and blue (-1 V)?

- It appears that for the competitive H₂ evolution reaction, the authors only considered the Volmer-Heyrovsky mechanism and ignored Volmer-Tafel. Volmer-Tafel is thought to dominate on Pt, for example. Were both mechanisms calculated? If there's reason to believe that Volmer-Tafel is inactive, please provide it.

- I found it confusing that the authors were including their results (for the limiting potential, U_{L}) already in their introduction, without telling us how they calculated these quantities or what they assumed in doing so.

- The authors spend a decent amount of time in the introduction saying why the CHE model is not good enough, yet then on page 8 they make claims about what the potential-determining steps are in N₂ and proton reduction, which as far as I can tell are based solely on CHE logic. (That is, the "last step to turn downhill".) Since they can potentially make a much better analysis with the CEP model, why are they relying on CHE here?

- In line 106, the authors cite 6 articles for the CEP model and explain it very briefly. There is no detailed description of the CEP model in the "methods" section as well. It is confusing to readers from which of these six articles the CEP model is coming from and how exactly it works. It is important to cite the original paper that developed the CEP model and explain in detail if authors used any modification of that method in this paper. Also, it is a good idea to discuss the method in the "method" section of the paper as it is the core of all calculations done; i.e., only the expert reader will recognize that the paragraph on "Implicit solvation...of the electrolyte" is referring to the CEP model.

- In Line 118 and elsewhere, ("Further calculations on other catalysts reveal that larger charge transfer..."). This is a general comment to the authors. There is no data available that is used to make plots in the manuscript. While charge transfer is discussed as an important parameter to decide the slope of ΔG vs U plot, there is no availability of the charge transfer values for any reaction in the manuscript or supporting information. It is important that all the data that is used to make any plot is available either in the form of mathematical equation (e.g. coverage) or tables (e.g. Gibbs free energy, charge transfer etc.).

- In microkinetic modeling, it is crucial to have a table specifying all rate constants and Gibbs free energies at all three pH and different potentials; or some way for the interested reader to reproduce the results.

- I found the comparison of CO₂RR with NRR to be confusing. It is unclear what the authors reasoning towards the premature decrease in NRR activity which is not seen for CO₂RR (when in competition with HER). In line 73, the authors suggest that the premature decrease in NRR activity should be attributed to the intrinsic properties of catalysts. However, in the rest of the manuscript, they do not mention what properties of the catalysts that could result in such a behavior. In fact, the rest of the manuscript only compares the elementary steps of NRR and HER to support their reasoning. I do not really understand what the CO₂RR comparison was doing for this manuscript.

- The authors use the term coverage (θ) for N₂ and H in the micro-kinetic modeling section. Also, in Figure 4, the y-axis represents a continuously varying coverage. However, several systems tested in this manuscript are single-site catalyst (for example Fe@N₄). How is coverage defined for such a system?

- How is pH incorporated when making the free energy surfaces? It is very clear that the bulk pH and local pH are very different. How do the authors treat the local pH? Also, based on my understanding the competition of NRR with HER should be more rigorous under acidic conditions provided the abundance of H⁺ to block the active sites. However, this should be less severe under alkaline conditions.

- The CHE definition in Supplementary Note 2 seems problematic. It is not the chemical potential of H⁺, but rather of the (H⁺ and e⁻) pair that can be related to the $\mu(\text{H}_2)$ at certain conditions. Also, credits should be given to the correct reference for CHE (i.e., Norskov 2004).

Reviewer #3 (Remarks to the Author):

The authors have written a very nice manuscript and should be commended for producing a very clean, concise, and well-reasoned work of science. As I outline below, the results of this paper are somewhat expected. The main contributions of this paper are 1. The insufficiency of the CHE model in describing the potential dependence of the species relevant to NRR and 2. Per microkinetic modeling too much importance is put on the limiting potentials from DFT. This work clearly outlines and corrects a flaw in the current thinking and literature around NRR electrocatalysis by testing the CHE model and introducing microkinetic modeling. I thus recommend this manuscript for publication with minor revisions.

Specific Comments:

The authors discuss the general trend of charge transfer being larger for H^{*} adsorption than N₂^{*} adsorption. While the demonstration of this trend is valuable, the trend itself is not surprising. N₂ is known to be very inert, and thus typically physisorbs on surfaces and thus has very limited charge transfer. Additionally, the greater dependence of H^{*} on applied potential relative to N₂^{*} is also expected. The adsorption energies of N₂^{*} and H^{*} are typically modeled using the computational hydrogen electrode model. In this model the adsorption of N₂ is assumed to be independent of applied potential, necessitating a crossover between H^{*} and N₂^{*} adsorption energies at some applied potential. The authors find this crossover as expected. While authors show in their manuscript that the CHE model is not sufficient to describe the real physics I present it here just to explain the expectations a reader would have. However, the differing potential dependence of N₂H^{*} relative to H^{*} as well as N₂^{*} having significant potential dependence at all is interesting and not expected. Much of this lengthy discussion is outlined in [1] below (reference 56 in the manuscript.) I am somewhat surprised this work is not referenced more extensively, as this piece clearly picks up the thread of this previous work.

The authors examine the three surface species that are important for NRR: N₂^{*}, N₂H^{*}, and H^{*}. However, there is a 4th important species: NH₂^{*}. NH₂^{*} is often seen control the other half of the volcano plot when scaling relations are generated [2]. Thus, the authors cannot claim to have fully examined the species relevant for NRR. This limitation should be noted.

The section titled "Origin and descriptor for different slopes" is an interesting discussion of why there are differing slopes in the adsorption vs potential of various species with respect to their extent of charge transfer. However, as the section lays out this is more a demonstration of the Nernst equation than a new discovery. This is not really a flaw, however it should not be posed as a major discovery.

[1] Singh, A. R. et al. Electrochemical Ammonia Synthesis-The Selectivity Challenge. ACS Catal. 7, 706-709 (2017).

[2] Computational Screening of Rutile Oxides for Electrochemical Ammonia Formation. (n.d.). doi:10.1021/acssuschemeng.7b02379.s001

Reviewer #1 (Remarks to the Author):

In the manuscript titled “Understanding Potential-dependent Competition Between Electrocatalytic Dinitrogen and Proton Reduction Reactions”, Choi et al. reported a computational study of nitrogen reduction reaction (NRR). By combining density functional theory (DFT) calculations and kinetic model, the authors concluded that the reason for the early loss of NRR efficiency attributes to the loss of N₂ adsorption when H adsorption becomes stronger at more negative applied potential. Thus, hydrogen evolution reaction (HER) becomes dominant. Overall, it looks to me that this manuscript is worth publishing. But it seems that the evidence currently provided by the authors cannot fully support their conclusions as they expected. Based on my expertise and understanding of NRR and HER. I raise several comments for the authors to consider as follows.

Reply: We are greatly grateful to the reviewer for positive comments on our work as well as very helpful suggestions. We respond to each question the reviewer brings one by one as below. The replies in this letter are marked in blue, while the revised texts in the manuscript and supporting information are highlighted in yellow.

1. I am more skeptical of the calculation results shown in Figure 5. According to the hypothesis proposed by the authors, competitive adsorption between N₂ and H* is the major cause of the competition between NRR and HER. More specifically, N₂ adsorption is less dependent on voltage. Therefore, the relationship between the adsorption energy and the voltage change is not obvious. Instead, the adsorption of H highly depends on voltage. Therefore, its adsorption energy has a significant correlation with voltage. The above analysis can explain the calculation results in Figure 4, but not Figure 5. According to the above assumptions, Figure 5 should also appear similar trend as shown in Figure 4. Although the

author emphasizes that CHE has limitations in describing charge transfer for transition states, the CHE model should capture the above-mentioned dependence of adsorption energy and voltage. I can't understand the significant differences between Figure 4 and Figure 5. I suggest the authors add more discussions and explanations.

Reply: We thank the reviewer for this insightful comment. As the reviewer points out, the H adsorption energy becomes more negative with more negative potential in both the CEP and CHE models. However, the key difference lies in the different slopes with which the energy (and activation energy) lowering occurs for Volmer reaction ($* + \text{H}_2\text{O} \rightarrow *^{\text{H}} + \text{OH}^-$) vs. Heyrovsky reaction ($*^{\text{H}} + \text{H}_2\text{O} \rightarrow \text{H}_{2(\text{g})} + \text{OH}^-$), which affects the net coverage of $*^{\text{H}}$. Specifically, in the CEP model in which the charge transfer is rigorously calculated, the charge transfer in the TS of Volmer reaction and that of the Heyrovsky reaction (e.g., activation energy slope) is 0.96 and 0.56, respectively. Also, $\Delta G(*^{\text{H}})$ decreases with the slope of 1.34, higher than that of the CHE model (1.0). Thus, with increasingly more negative U , the rate for the Volmer reaction ($*^{\text{H}}$ formation) becomes faster than that for the Heyrovsky reaction ($*^{\text{H}}$ elimination), which leads to the potential-dependent accumulation of the $*^{\text{H}}$ coverage.

However, in the CHE model, since the same potential-dependence of the reaction energetics is assumed as long as the reactions involve the same number of electrons, both Volmer ($*^{\text{H}}$ formation) and Heyrovsky ($*^{\text{H}}$ elimination) reactions formally involving 1-e transfer, have the same potential-dependent behaviors. As a result, the accumulation of the H coverage is not observed with potential. To verify it, we plotted the MKM results using the CHE model for a hypothetical case (inspired by the CEP results) in which the activation energy of Volmer and that of Heyrovsky reactions have different slopes of 0.9 and 0.5, respectively, as well as the identical slopes of 0.5 (a CHE case shown in the original manuscript). As the comparison is shown in Supplementary Fig 16 and below, due to the identical potential-dependence

assumed in CHE model, the accumulation of H coverage with U is not observed in usual treatment of the same slopes, and the H-coverage crossover with potential is only observed in a hypothetical case in which the slopes for Volmer and Heyrovsky reactions are different, a situation that only the CEP model can treat rigorously.

We now added such discussions in the main text (in Page 19) with Supplementary Fig. 16.

Supplementary Fig. 16 | The change in θ_{N_2} , θ_H and r_{NH_3} by U obtained by the microkinetic modeling (MKM) using the CHE model with different charge transfer in TS of $*H$ formation. The relative r_{NH_3} and coverage are shown in upper and lower panels, respectively. Electrode potential (U) is in RHE scale at pH = 13.

2. What is the physical meaning of dG/U in Figure 6? Is it a capacitance?

Reply: The slope ($\Delta G/U$) is change of ΔG with respect to the change of potential and its physical meaning (ΔG (in eV) / ΔU (in V)) is the amount of transferred electrons (e^-) during chemical reaction.

3. I cannot agree with the authors in the discussions of Figure 1. Both of the above reactions

are multiple-step reactions (NRR is 6 e- and HER is 2e-). The overall reduction potential, in most cases, is known to be not related to the performance. Therefore, I cannot be convinced that “NRR should have more negative potential than CO₂RR” as the authors concluded.

Reply: We thank and agree with the reviewer’s comment. We now revised the discussion of Fig. 1 as below.

“The experimental CO₂RR activity increases with more negative potential and the maximum CO₂RR activity is observed at around -0.7 V (Fig. 1b)²⁴. At $U = -0.7$ V which is more negative than the theoretical limiting potential U_L (-0.32 V), CO₂RR can be sufficiently facilitated and its activity begins to decrease due to approaching the mass-transfer limit. Thus, the potential-dependent CO₂RR activity can be qualitatively explained by conventional DFT calculations. NRR activity also increases with more negative potential at first, however, it begins to decrease quickly at -0.4 V (pH = 7.2) or -0.05 V (pH = 13) (Fig. 1b), much earlier before reaching its U_L (-1.29 V). This result indicates that NRR activity prematurely decreases with increasing reduction potential while its kinetics has not reached its expected theoretically maximum. NRR shows an unusual potential-dependent behavior which is unexplained by the conventional DFT calculations. Thus the premature decrease in the NRR activity should be attributed to the intrinsic properties of catalysts.”

Reviewer #2 (Remarks to the Author):

Choi et al used electronic structure calculations in order to understand the competition between electrochemical ammonia production and hydrogen evolution on a single-site iron catalyst. This is an interesting and important research topic and I believe the use of grand-canonical DFT methods can provide significant new insights to processes like these. Unfortunately, I found this work to suffer from many severe flaws (discussed below) and to not provide meaningful insights to the literature.

Reply: We are greatly grateful to the reviewer for many very helpful suggestions to enrich the perspectives of our paper. As per the reviewer's suggestions, we included the activation energies of all the other reaction steps in the MKM, and extensively clarified the physical origin for the large difference between the CEP and CHE models. We respond to each question the reviewer brings one by one as below. The replies in this letter are marked in blue, while the revised texts in the manuscript and supporting information are highlighted in yellow.

Major flaws:

1. The free energy diagrams found in Figure SI-3 form the basis for most of this manuscript. Here, they compare the diagrams produced from the well-known computational hydrogen electrode (CHE) model to what they are referring to as the constant-electrode potential (CEP) model. It is shocking how different the results are between the two models, especially since the CHE model is thermodynamically exact in the limit of integer electron transfer (or alternatively, of zero change in surface-normal dipole). If their CEP model is thermodynamically consistent, I would expect deviations from the CHE model to be small, perhaps on the order of 0.1 eV (for example, if an electron transfer was 0.9 instead of 1). Yet

we can observe dramatic differences: for example, the step $*\text{NHNH} \rightarrow *\text{NHNH}_2$ appears uphill by ~ 0.5 eV in the CHE model and downhill by about the same amount in the CEP model. This implies that CHE made a 1-electron error in this step! This is hard to believe; if this is really the case, the authors need to fully explain the difference and the physical meaning of such.

Reply: We thank the reviewer for this comment. The reaction energy of $*\text{NHNH} + (\text{H}^+ + \text{e}^-) \rightarrow *\text{NHNH}_2$ the reviewer is referring to, we are guessing, is -0.30 eV in the CEP model and 0.33 eV in the CHE model (instead of 0.5 eV). The physical origin for such a large discrepancy (0.63 eV) is the change in the potential of zero charge (U_{PZC}) during reaction. The U_{PZC} represents the electrode potential of the slab model in neutral state (net charge = 0). For example, the U_{PZC} of Fe@N_4 (denoted as $U_{\text{PZC}}(*)$) is -0.83 V and that of $*\text{H}$ ($U_{\text{PZC}}(*\text{H})$) is -0.54 V. To set the U of $*$ and $*\text{H}$ to 0 V, electrons amounting to 0.83 V is extracted in $*$ while electrons amounting to 0.54 V is extracted in $*\text{H}$. Consequently, extra electrons corresponding to 0.29 V ($0.83 - 0.54$) are engaged to compensate the change of U_{PZC} . In the CHE model, all intermediates are assumed to be at same U , and thus, the charge due to the change of U_{PZC} is not included. We compared the change in U_{PZC} (ΔU_{PZC}) with $\Delta\Delta G_{\text{CEP-CHE}}$. Here, ΔU_{PZC} and $\Delta\Delta G_{\text{CEP-CHE}}$ represents the change in U_{PZC} by adsorption (e.g. $(\Delta U_{\text{PZC}}(*\text{NH}_2) = U_{\text{PZC}}(*\text{NH}_2) - \Delta U_{\text{PZC}}(*))$ and $(\Delta G$ obtained by the CEP model $- \Delta G$ obtained by the CHE model at 0 V vs. SHE), respectively.

A linear correlation of ΔU_{PZC} with $\Delta\Delta G_{\text{CEP-CHE}}$ was obtained, indicating that the discrepancy mainly originates from the change in U_{PZC} during reaction. We found that $\Delta U_{\text{PZC}}(*\text{NHNH})$ is -0.03 V, while $\Delta U_{\text{PZC}}(*\text{NHNH}_2)$ is -0.26 V. Thus, the $\Delta\Delta G_{\text{CEP-CHE}}(*\text{NHNH})$ (-0.09 eV) is much closer to 0 than $\Delta\Delta G_{\text{CEP-CHE}}(*\text{NHNH}_2)$ (-0.71 eV), leading to the large discrepancy in $*\text{NHNH} \rightarrow *\text{NHNH}_2$ step between the CEP model and CHE model.

Incidentally, we note that similar approximately 0.6 eV of difference in the CEP vs. CHE models has been reported in the literatures (reference [1] and [2] in below). For example, the ΔG of $*N + (H^+ + e^-) \rightarrow *NH$ on Ru single atom catalyst is 0.26 eV in the CHE model, while that of the CEP model is -0.41 eV at 0 V [1].

These physical explanations are now added in the main text (in Page 13) as well as Supplementary Note 4 with Supplementary Fig. 11.

[1] Ruthenium single-atom catalysis for electrocatalytic nitrogen reduction unveiled by grand canonical density functional theory. *J. Mater. Chem. A* 2020, 8, 20402

[2] Substantial potential effects on single-atom catalysts for the oxygen evolution reaction simulated via a fixed-potential method. *J. Catal.* 2020, 391, 530

Supplementary Fig. 11 | Change of U_{PZC} by adsorption and $\Delta\Delta G_{CEP-CHE}$. (a) U_{PZC} vs. SHE (V) of each adsorbate. Horizontal red dashed line represents U_{PZC} of vacant Fe@N₄ site ($U_{PZC}(*)$). (b) Relationship between ΔU_{PZC} and $\Delta\Delta G_{CEP-CHE}$. The $\Delta\Delta G_{CEP-CHE}$ is obtained by ΔG obtained by the CEP model - ΔG obtained by the CHE model at 0 V (vs. SHE).

2. The mechanism they have employed is not justified. There are many possible pathways that this reaction could follow, even on a single-site catalyst, and the authors have only listed

one, without justification.

Reply: We thank the reviewer for this highly relevant and helpful comment to enrich our results. We now considered all possible reaction intermediates listed in Fig. 2. Combined with the Comment 7, we added activation free energies. The lowest activation energy pathway is highlighted with red line in Fig. 3. Consideration of these additional energetics of intermediates do not change the conclusions.

Fig. 2 | Calculation models for Fe@N₄ catalysts. Fe@N₄ with (a) a hexagonal ice bilayer water and (b) a hexagonal ice bilayer water containing a solvated H₃O⁺. Top-view and side-

view are shown in the upper panel and lower panel, respectively. (c) The optimized geometries of all possible reaction intermediates of NRR. The number of transferred protons is listed in the first row. For $*NH + NH_3$, $*NH_2 + NH_3$, $*NH_3 + NH_3$, NH_3 is omitted for the clarity.

Fig. 3 | Free energy diagram of NRR including activation energy. Free energy diagram of NRR (pH = 13) at (a) $U = 0$ V vs. RHE, (b) $U = -0.23$ V vs. RHE and (c) $U = -0.5$ V vs. RHE. The lowest activation energy requiring reaction pathway is represented by red line.

3. Related, the free energy diagrams have a step that appears to liberate NH_2 ($*NH_2NH_2 \rightarrow *NH_2 + NH_2$). This is a free radical! This seems very unlikely. Why was only this single mechanism explored? There are many ways that N-N bond could be broken, and I do not expect the liberation of a radical from the surface would be the most favorable way. (Additionally, they calculate the liberation of this radical to be exergonic, which seems suspect.)

Reply: We thank the reviewer for pointing this out. The “ $*NH_2 + NH_2$ ” was a typo. The correct mechanism which we considered is $*NH_2NH_2 + (H^+ + e^-) \rightarrow *NH_2 + NH_3$, liberating NH_3 molecule. We now corrected the typos in all free energy diagrams.

4. In Figure 2, the transition states corresponding to part (c), (e) and (f) look quite similar to the final states. I.e., I do not see any significantly stretched or distorted bonds. How do the authors confirm that TS obtained are the true transition states? For clarity and better

understanding, the authors should include the charge/electron transfer associated between the IS, TS and FS, as well as potential-energy diagrams of the minimum energy pathway.

Reply: We verified the TS to be a first order saddle point by performing vibrational analysis. The TS structure of *NNH formation is especially difficult to visually differentiate from its FS since the *NNH formation is highly endothermic, and hence, the geometry as well as charge transfer in the TS are very similar to those of FS. This agrees with the Hammond postulate [3], that is, TS of endothermic reaction resembles FS (and vice-versa). We now added the discussion for the geometry of TS (in Page 12), images for reaction pathway (Fig. 4), charge transfer and energy diagram (Supplementary Fig. 4, 6 and Supplementary Table 4) for the clarity.

[3] A Correlation of Reaction Rates. *J. Am. Chem. Soc.* 1955, 77, 334

Fig. 4 | Reaction pathway of $*H$ formation and $*NNH$ formation. Reaction pathway for (a) $*N_2 + H_2O \rightarrow *NNH + OH^-$ and (b) $* + H_2O \rightarrow *H + OH^-$. Side-view and top-view are listed in the upper panel and lower panel, respectively. Transition state of each reaction is highlighted with blue dashed box. Green and purple balls represent the transferred H atoms during reorganization.

Supplementary Table 4 | Data in Fig. 4b. The amount of charge transfer in TS/FS is in parenthesis.

U (V)	$\Delta G_a(*H)$ [H ₃ O ⁺]	$\Delta G_a(*N_2 \rightarrow *NNH)$ [H ₃ O ⁺]	$\Delta G_a(*H)$ [H ₂ O]	$\Delta G_a(*N_2 \rightarrow *NNH)$ [H ₂ O]
0	0.74 (0.78 / 1.14)	1.15 (0.30 ^a) / 0.31)		
-0.25	0.55 (0.72 / 1.12)	1.09 (0.32 ^a) / 0.32)		
-0.5	0.37 (0.69 / 1.05)	0.98 (0.53 ^a) / 0.53)		
-0.77			0.79 (0.99 / 1.19)	1.50 (0.72 / 0.96)
-1	-0.03 (0.84 / 1.22)	0.65 (0.79 / 0.90)	0.55 (1.02 / 1.35)	1.32 (0.76 / 1.12)
-1.27			0.31 (0.67 / 1.19)	1.11 (0.75 / 1.37)

^a) Charge transfer in TS is almost same with that of FS.

5. If I understand it correctly, Figure 3(a) is pure thermodynamics and should align with CHE in the limit of integer electron transfer; that is, deviations from integer slope should be caused by non-integer electron transfer. To what do the authors attribute a ~0.2 e transfer for *N₂ adsorption? I would have suspected that this elementary step doesn't involve significant charge transfer. Similarly, why are the charge transfers associated with the other two steps not closer to 1? [As a more minor point, how are the slopes calculated? The data points are not connected by a single straight line.]

Reply: As in the response to the Comment 1, non-integer electron transfer in the CEP model originates from the change of U_{PZC} during reaction. Thus, we compared the deviation of the charge transfer with ΔU_{PZC} for *H and all reaction intermediates of NRR. Here, the deviation of the charge transfer is obtained by the difference between the slopes obtained by the CEP model (non-integer) and that of CHE model (integer). We found that the deviation of the charge transfer well agrees with ΔU_{PZC} (Supplementary Fig. 12 in below). This result

indicates that electrochemical reactions which induce larger change of U_{PZC} , requires extra (or deficient) electrons.

The physical origin of charge transfer in N_2 adsorption is π back-bonding, which is the most important mechanism for N_2 binding at transition metal atoms ([4], [5]). The back-donation of metal d electrons to LUMO state of N_2 (antibonding π^*) can significantly weaken the N-N triple bond and activates N_2 . Thus, the amount of charge transfer from metal to $*N_2$ is an important descriptor for estimating the extent of the N_2 activation [6]. We found that the transferred charge to $*N_2$ and N-N bond length increase with more negative U (Supplementary Fig. 6 in below). This result indicates that more negative potential promotes charge transfer to $*N_2$ and $*N_2$ activation.

Due to the geometry change by U (which will be discussed during the response to Comment 10 below), the data points of $*N_2$ and $*NNH$ are not completely in a single line in a wide potential range (0 V \sim -1 V). The slopes are obtained by a simple first order linear regression. In Fig. 3a (Fig. 5a in the revised manuscript), we focused on the trend in ΔG vs. U rather than comparing the exact slopes. We note that more narrow range of U (e.g. 0 V \sim -0.5 V) is used to calculate accurate slope to construct the potential-dependent energetics in the MKM.

These discussions are now added in Page 10-13 with Supplementary Fig. 5 and 12.

[4] Intrinsic Dinitrogen Activation at Bare Metal Atoms. *Angew. Chem. Int. Ed.* 2006, 45, 6264

[5] Design of a Metal–Organic Framework with Enhanced Back Bonding for Separation of N_2 and CH_4 . *J. Am. Chem. Soc.* 2014, 136, 698

[6] Jahn–Teller Distorted Effects To Promote Nitrogen Reduction over Keggin-Type Phosphotungstic Acid Catalysts: Insight from Density Functional Theory Calculations. *Inorg. Chem.* 2019, 58, 7852

Supplementary Fig. 12 | Relation of ΔU_{PZC} with slope on Fe@N₄.

Supplementary Fig. 5 | Net charge on *N₂ and N-N bond length with U. The net charge in *N₂ and N-N bond length are represented by black line (left) and red line (right), respectively. The net charge is obtained by the Bader charge analysis.

6. Similarly, Figure 6 does not make sense to me. (The authors are plotting nearly the same quantity in the x and y directions.) It is bizarre to see a huge window in the charge transfer associated with *N₂ adsorption (ranging from 0 to almost 0.6 e). How do the authors justify a charge transfer of 0.6 e for a step that doesn't involve explicit charge transfer? Similarly, for *H adsorption, the charge-transfer window varies over a huge window from roughly 1 to 2 e; how could the authors justify the 2-electron case (almost equivalent to the total electrons transferred in the entire HER)?

Reply: Again, the deviation of charge transfer mainly originates from the change of U_{PZC} (ΔU_{PZC}). Thus, we compared the deviation of the charge transfer with ΔU_{PZC} for $\Delta G(*H)$, $\Delta G(*N_2)$ and $\Delta G(*N_2 \rightarrow *NNH)$ on various catalysts. We found that the deviation of the charge transfer well agrees with ΔU_{PZC} (Supplementary Fig. 19 in below). This result

indicates that catalysts whose U_{PZC} easily changes during chemical reaction, requires extra (or deficient) electrons during electrochemical reaction.

We also note that such a large deviation in charge transfer has also been reported on SAC and N-doped graphene ([7], [8]), which are incorporated in our system (Fe@N₄). For electrochemical CO₂ reduction, the slopes for the CO₂ adsorption and *COOH formation are 0 and 1, respectively in the CHE model. However, on Fe@N₄, the slopes for CO₂ adsorption and *COOH formation are reported as ~1 and ~1.64, respectively [7]. In case of ORR on N-doped graphene, the slopes ranging from 0.24 to 1.71 were reported on N-doped graphene, significantly deviated from 1 in the CHE model [8]. Thus, our results combined with the literatures coherently point to a conclusion that such a large charge transfer at N-doped graphene based SACs can be obtained when U_{PZC} easily changes during reaction. Related discussions are now added in the “Origin and descriptor for different slopes” section and Supplementary Fig. 19.

[7] Dipole-Field Interactions Determine the CO₂ Reduction Activity of 2D Fe–N–C Single-Atom Catalysts. *ACS Catal.* 2020, 10, 7826

[8] Evaluating Potential Catalytic Active Sites on Nitrogen-Doped Graphene for the Oxygen Reduction Reaction: An Approach Based on Constant Electrode Potential Density Functional Theory Calculations. *J. Phys. Chem. C* 2020, 124, 25675

Supplementary Fig. 19 | Linear correlation between the slope and ΔU_{PZC} . Black, blue and red colors represent $\Delta G(*H)$, $\Delta G(*N_2)$ and $\Delta G(*N_2 \rightarrow *NNH)$, respectively.

7. The microkinetic results seem are suspect, since they didn't calculate barriers for many of their steps. (It seems they only have barriers for the couple of steps that they declared to be "potential determining" in the manuscript; I do not see others reported.) I personally would say that all barriers should be included in such a model since they are now rather straightforward with constant-potential methodologies; although I agree that it's likely that setting some to zero won't affect the kinetics. However, I can see a couple of apparently significant omissions just from looking at the free energy diagrams: * At 0 V, the highest-energy intermediate is $*NHNH_2$ or $*NHNH$, depending on if you examine the CHE or CEP model. (As noted earlier, the difference in elementary thermodynamics between these two approaches is troubling.) Therefore, it's a good guess that the largest *overall* barrier (measured from the most-abundant reactive intermediate, MARI) would be from either the

formation or destruction of these high-energy species. The largest *overall* barrier, not the largest individual barrier, is what typically best approximates the kinetics, so the omission of these barriers is significant. * There are non-electrochemical steps involved, and notably $*\text{NH}_2\text{NH}_2 \rightarrow *\text{NH}_2 + \text{NH}_2$. Since this step will not be affected by potential (to a first order), its barrier will not diminish with potential as the researchers are implicitly assuming other steps will. Therefore, this barrier and ones like it must be calculated. It is a bit strange to examine N_2 reduction without examining breaking of the N-N bond.

Reply: We thank the reviewer for helpful suggestions to strengthen our results. Except for the adsorption/desorption of N_2 and NH_3 , activation energies are now calculated for all reaction steps when using H_2O as a proton donor. We note that highly exothermic reactions such as $*\text{NH} + \text{H}_2\text{O} \rightarrow *\text{NH}_2 + \text{OH}^-$ and $*\text{NH}_2 + \text{H}_2\text{O} \rightarrow *\text{NH}_3 + \text{OH}^-$ proceed barrierlessly. In the lowest activation energy pathway at 0 V (red line in Fig. 3a below), the $*\text{NHNH}$ formation has a higher apparent activation energy than the $*\text{NNH}$ formation as in the reviewer's expectation. Thus, we now included the activation energies of all reaction steps in the MKM simulations under neutral and alkaline conditions. The revised MKM results at pH 7.2 and 13 are shown in Fig. 6 as below.

Since the apparent activation of $*\text{NNH}$ formation decreases slower than that of $*\text{NHNH}$ formation with more negative U , the $*\text{NNH}$ formation begins to show the highest apparent activation energy (i.e. RDS) at -0.12 V (Fig. 3b and 3c). In this study, we mainly discuss NRR under pH = 13 and pH = 7.2, the same conditions with the reported experiments. Thus, we simplified the MKM simulations under acidic conditions by considering only $*\text{NNH}$ formation in activation energy, assuming that $*\text{NNH}$ formation determines overall kinetics of NRR under negative potential. At all three conditions considered in this study, the premature decrease of ammonia yield rate and N_2 coverage still hold in the revised MKM results (Fig.

6a) with the precise U at maximum NH_3 yield rate or N_2 coverage changing only slightly by ~ 0.05 V.

Fig. 3 | Free energy diagram of NRR including activation energy. Free energy diagram of NRR (pH = 13) at (a) $U = 0$ V vs. RHE, (b) $U = -0.23$ V vs. RHE and (c) $U = -0.5$ V vs. RHE. The lowest activation energy requiring reaction pathway is represented by red line.

Fig. 6 | The change in θ_{N_2} , θ_H and r_{NH_3} by U obtained by the microkinetic modeling (MKM) using the CEP model or CHE model. (a) MKM results using the CEP model and (b) MKM results using the CHE model at three different pH (pH = 13, pH = 7.2 and pH = 0). The relative r_{NH_3} is obtained by dividing the r_{NH_3} by its maximum. Dashed lines represent MKM results without HER. The relative r_{NH_3} and coverage are shown in upper and lower panels, respectively.

8. They seem to give two different explanations for the H_2 production overwhelming N_2 reduction (which is advertised to be the significant conclusion of this manuscript). On page 8, they attribute it to the higher coverage of H as the potential gets more negative (apparently blocking sites for N_2 adsorption). Later on the same page they seem to be making the argument that it is the barrier to form $*NNH$, as compared to the barrier for $*H$, that limits.

Which is it? I don't expect two different steps to be equally responsible.

Reply: The coverage crossover between H and N₂ is mainly responsible for the premature decrease of NRR activity (i.e. decreasing NRR activity with more negative U). However, higher *NNH formation barrier than *H formation explains why the NRR kinetics is slower than HER in absolute magnitudes. We now clarified the role of coverage crossover and *NNH formation vs. *H formation.

Significant issues:

There are many other significant issues with this manuscript, which are too numerous to mention. I will list some of them here, but I do not mean to imply this is a complete list.

9. The free energy diagrams should be in the manuscript itself; it's too confusing to have to look to ill-explained SI figures to follow the logic of the discussion in the manuscript.

Reply: We now moved the potential-dependent free energy diagram of NRR at $U = 0, -0.23$ and -0.5 V (vs. RHE in pH 13) to the main text as Fig. 3.

10. In Supplementary figure 6, why is the red plot (-0.5 V) not half-way between the black (0 V) and blue (-1 V)?

Reply: The reason for that the reaction energy at -0.5 V is not half-way between that at 0 V and -1 V, is the change of *NNH geometry with U . We optimized all structures with the proper number of electrons in the slab model and found that *NNH optimizes to a more bent structure with negative potential. As a comparison, we also obtained the potential-dependent *NNH formation energy when the geometry of * and *NNH is fixed at their optimized geometries at neutral state. Without the needed potential-dependent geometry relaxation, the

slope (ΔG vs. U) is constant (Supplementary Fig. 7 in below). This result indicates that the geometry relaxation by U leads to the potential-dependent charge transfer. We now added such discussion in the Page 12-13 with Supplementary Fig. 7 as below.

Supplementary Fig. 7 | The potential-dependent change of geometry and formation energy of *NNH. (a) The optimized geometry of *NNH at neutral state (-0.7 V), 0 V, -0.25 V, -0.5 V and -1 V (vs. SHE). (b) Potential-dependent *NNH formation energy. Red line represents formation energy obtained by single point calculation using optimized structure at neutral state, while black line represents *NNH formation energy at fully optimized structure. At -0.5 V to -1 V, noticeable changes of geometry and *NNH formation energy are not observed.

11. It appears that for the competitive H_2 evolution reaction, the authors only considered the Volmer-Heyrovsky mechanism and ignored Volmer-Tafel. Volmer-Tafel is thought to dominate on Pt, for example. Were both mechanisms calculated? If there's reason to believe that Volmer-Tafel is inactive, please provide it.

Reply: The Tafel reaction can proceed with two different pathways as illustrated in Supplementary Fig. 13. One is the reaction of H at Fe (*H) with H at adjacent N atom (*-NH) and the other is the reaction of two H's at Fe atom (*2H). We first compared the formation free energy of *-NH and *H by using the CEP model and found that *H is more stable than

*-NH by 0.92 eV (Supplementary Fig. 14a). Thus, we exclude adjacent N atom as an active site of HER. Next, we compared the activation energy of *2H formation ($*\text{H} + \text{H}_2\text{O} \rightarrow *2\text{H} + \text{OH}^-$) and that of Heyrovsky reaction ($*\text{H} + \text{H}_2\text{O} \rightarrow \text{H}_2(\text{g}) + \text{OH}^-$) (Supplementary Fig. 14b). The activation energy for *2H formation is higher than that of Heyrovsky reaction by 0.34 eV at 0 V. Thus, we conclude that the Volmer-Tafel mechanism is less active, and thus considered the Volmer-Heyrovsky reaction. Above discussion are now added in the Supplementary Note 5 with Supplementary Fig. 14-15.

Supplementary Fig. 14 | Possible reaction pathways for HER on Fe@N₄. Grey, brown, blue and white balls represent Fe, C, N and H atom, respectively. *H and *-NH represents H adsorbed on Fe and adjacent N atom to Fe, respectively.

Supplementary Fig. 15 | Comparison of the Volmer-Heyrovsky pathway with the Volmer-Tafel pathway. (a) Formation free energy for $*H$ and $*-NH$. (b) Activation free energy for $*2H$ formation ($*H + H_2O \rightarrow *2H + OH^-$) and Heyrovsky reaction ($*H + H_2O \rightarrow H_2(g) + OH^-$). All energies are represented at 0 V vs. RHE under pH = 13. Grey, brown, blue, red and white balls represent Fe, C, N, O and H atom, respectively.

12. I found it confusing that the authors were including their results (for the limiting potential, U_L) already in their introduction, without telling us how they calculated these quantities or what they assumed in doing so.

Reply: The limiting potential (U_L) represents electrode potential where all individual electrochemical reactions become exothermic. In the CHE model, U_L is obtained from the free energy change at potential determining step (denoted as ΔG_{PDS}) as $U_L = -\Delta G_{PDS} / e$ (V). We now added the definition of the limiting potential in the introduction as below.

“Here, we calculated U_L by the computational hydrogen electrode (CHE) model, which has been the most widely used method in estimating the energetics of electrochemical reactions. The U_L is equal to the $\Delta G_{PDS} / e$, where ΔG_{PDS} is the free energy change at the most uphill individual step (i.e. potential-determining step).”

13. The authors spend a decent amount of time in the introduction saying why the CHE model is not good enough, yet then on page 8 they make claims about what the potential-determining steps are in N₂ and proton reduction, which as far as I can tell are based solely on CHE logic. (That is, the "last step to turn downhill".) Since they can potentially make a much better analysis with the CEP model, why are they relying on CHE here?

Reply: The PDS is the highest overpotential requiring individual step among electrochemical reactions, which determines the overall thermodynamic overpotential. Thus, the PDS is the same in the CHE model and CEP model for both NRR and HER. To clarify, we now modified the sentence to “Next, we analyze the trend of NRR and HER activity by comparing $G_a(*N_2 \rightarrow *NNH)$ and $G_a(*H)$, a RDS under negative U as well as PDS in NRR and HER on Fe@N₄ (Fig. 3 and Supplementary Fig. 4), respectively.”

14. In line 106, the authors cite 6 articles for the CEP model and explain it very briefly. There is no detailed description of the CEP model in the “methods” section as well. It is confusing to readers from which of these six articles the CEP model is coming from and how exactly it works. It is important to cite the original paper that developed the CEP model and explain in detail if authors used any modification of that method in this paper. Also, it is a good idea to discuss the method in the "method" section of the paper as it is the core of all calculations done; i.e., only the expert reader will recognize that the paragraph on "Implicit solvation...of the electrolyte" is referring to the CEP model.

Reply: As per the reviewer’s suggestion, we now clarified the explanation of the CEP model used here in the Introduction as well as method sections.

In the Introduction, “In this work, we use the constant electrode potential (CEP) model which treats the electrode-electrolyte interface as a polarizable continuum with implicit solvation model^{34,35}. In this model, the number of electrons is adjusted to guarantee different states to have the same workfunction in the grand canonical states. This method has recently been used to understand many electrochemical reactions³⁶⁻⁴⁶.”

In the method section, “We treat the electrode-electrolyte interface as a polarizable continuum via the linearized Poisson-Boltzmann equation which is implemented in the VASPsol^{34,35}, extension of VASP. In this method, an ionic counter-charges are implicitly placed at the interface. The net dipole originates from adding (or extracting) electrons is screened by the ionic counter-charges.”

15. In Line 118 and elsewhere, (“Further calculations on other catalysts reveal that larger charge transfer...”). This is a general comment to the authors. There is no data available that is used to make plots in the manuscript. While charge transfer is discussed as an important parameter to decide the slope of ΔG vs U plot, there is no availability of the charge transfer values for any reaction in the manuscript or supporting information. It is important that all the data that is used to make any plot is available either in the form of mathematical equation (e.g. coverage) or tables (e.g. Gibbs free energy, charge transfer etc.).

Reply: The charge transfer at each potential and the slope are now added as Supplementary Table 6. All energies and charge transfer related to the figures in manuscript were now listed in Supporting Information. The mathematical equation for obtaining the coverage is also added in the method of MKM section (Supplementary Note 6).

Supplementary Table 6 | The ΔN_e and slope for $\Delta G(*H)$, $\Delta G(*N_2)$, $\Delta G(*N_2 \rightarrow *NNH)$ and $\Delta G(*NH_2 \rightarrow NH_3)$. The electrode potential is in V vs. SHE.

Reactions	U (V)	Fe(110)	Ru(0001)	Rh(111)	Fe@N ₃	Fe@N ₄	Ru@NC ₂	Ru@N ₃	Ag@N ₄
	0	1.03	1.04	1.04	1.89	1.27	1.50	1.46	1.72
* + H ⁺	-0.5	1.01	1.00	1.01	1.88	1.10	1.47	1.63	1.68
→ *H	-1	0.99	0.97	0.99	1.85	1.28	1.46	1.58	1.79
	slope	1.12	0.98	0.99	1.92	1.15	1.46	1.57	1.75
	0	0.02	-0.05	0.00	0.31	0.34	0.47	0.32	0.02
* + N ₂	-0.5	-0.03	-0.03	-0.02	0.79	0.10	0.58	0.22	0.04
→ *N ₂	-1	-0.03	0.00	-0.01	0.81	0.28	0.35	0.22	-0.02
	slope	0.05	-0.06	-0.05	0.70	0.16	0.50	0.21	0.02
	0	0.78	0.80	0.70	1.14	0.34	0.83	1.00	0.96
*N ₂ + H ⁺	-0.5	0.82	0.83	0.74	0.98	0.91	0.39	1.57	0.97
→ *NNH	-1	0.85	0.85	0.79	1.00	0.89	1.13	1.56	1.25
	slope	0.82	0.84	0.74	1.07	0.66	0.62	1.39	1.03

16. In microkinetic modeling, it is crucial to have a table specifying all rate constants and Gibbs free energies at all three pH and different potentials; or some way for the interested reader to reproduce the results.

Reply: The potential-dependent energetics at all three pH used for the MKM, are now listed in the Supplementary Table 2. The formation energy (or activation energy) at certain U is expressed as $\Delta G(U)$ (or $G_a(U)$) = $\alpha U + \beta$, where the α and β represents the slope (or the amount of charge transfer in the reaction) and ΔG (or G_a) at 0 V, respectively. Thus, α and β of all reactions included in the MKM are provided. We used the transition state theory, thus the rate constants can be derived with just Gibbs free energy of activation.

Supplementary Table 2 | List of the potential-dependent energetics at different pH. The α and β are represented in the RHE scale at each pH value. Activation energy for adsorption/desorption of N₂ and NH₃ is not considered. The unit of β is eV.

Reactions	Alkaline (pH = 13)		Neutral (pH = 7.2)		Acidic (pH = 0)	
	α	β	α	β	α	β
$G_a(*H)$	0.96	1.49	0.96	1.11	0.72	0.73
$\Delta G(*H)$	1.34	0.44	1.34	0.56	1.12	0.63
$G_a(*H \rightarrow H_{2(g)})$	0.56	0.39	0.56	0.59	0.43	0.26

$\Delta G(\text{H}_{2(\text{g})})$	2	0	2	0	2	0
$\Delta G(*\text{N}_2)$	0.30	0.31	0.30	0.41	0.18	0.40
$G_{\text{a}}(*\text{N}_2 \rightarrow *\text{NNH})$	0.76	1.50	0.76	1.76	0.33	1.16
$\Delta G(*\text{NNH})$	1.40	1.66	1.40	1.80	0.33	1.44
$G_{\text{a}}(*\text{NNH} \rightarrow *\text{NNH}_2)$	0.54	0.26	0.54	0.44		
$\Delta G(*\text{NNH}_2)$	2.06	1.88	2.06	1.90	1.74	1.63
$G_{\text{a}}(*\text{NNH}_2 \rightarrow \text{N} + \text{NH}_{3(\text{aq})})$	0.72	0.41	0.72	0.66		
$\Delta G(*\text{N})$	0.43	2.06	0.43	2.21	-0.06	2.06
$G_{\text{a}}(*\text{N} \rightarrow *\text{NH})^{\text{a)}}$						
$\Delta G(*\text{NH})$	1.57	1.54	1.57	1.74	1.18	1.80
$G_{\text{a}}(*\text{NH} \rightarrow \text{NH}_2)^{\text{a)}}$						
$\Delta G(*\text{NH}_2)$	2.25	0.48	2.25	0.57	2.10	0.62
$G_{\text{a}}(*\text{NH}_2 \rightarrow *\text{NH}_3)^{\text{a)}}$						
$\Delta G(*\text{NH}_3)$	2.83	0.05	2.83	-0.01	2.59	-0.51
$G_{\text{a}}(*\text{NNH} \rightarrow *\text{NHNH})$	0.53	0.25	0.53	0.44		
$\Delta G(*\text{NHNH})$	2.09	1.58	2.09	1.61	1.86	1.69
$G_{\text{a}}(*\text{NHNH} \rightarrow *\text{NHNH}_2)$	0.24	0.29	0.24	0.37		
$\Delta G(*\text{NHNH}_2)$	3.03	2.01	3.03	2.02	2.28	1.38
$G_{\text{a}}(*\text{NHNH}_2 \rightarrow *\text{NH}_2\text{NH}_2)^{\text{a)}}$						
$\Delta G(*\text{NH}_2\text{NH}_2)$	3.83	1.63	3.83	1.57	3.63	1.28
$G_{\text{a}}(*\text{NH}_2\text{NH}_2 \rightarrow *\text{NH}_2 + \text{NH}_{3(\text{aq})})^{\text{a)}}$						
$G_{\text{a}}(*\text{NNH}_2 \rightarrow *\text{NHNH}_2)$	0.31	0.12	0.31	0.23		
$G_{\text{a}}(*\text{NHNH}_2 \rightarrow *\text{NH} + \text{NH}_{3(\text{aq})})^{\text{a)}}$						
$\Delta G(\text{NH}_{3(\text{aq})})$	3	-0.31	3	-0.31	3	-0.31

^{a)} These reactions proceed without barrier.

17. I found the comparison of CO₂RR with NRR to be confusing. It is unclear what the authors reasoning towards the premature decrease in NRR activity which is not seen for CO₂RR (when in competition with HER). In line 73, the authors suggest that the premature decrease in NRR activity should be attributed to the intrinsic properties of catalysts. However, in the rest of the manuscript, they do not mention what properties of the catalysts that could

result in such a behavior. In fact, the rest of the manuscript only compares the elementary steps of NRR and HER to support their reasoning. I do not really understand what the CO₂RR comparison was doing for this manuscript.

Reply: The main motivation of the comparison of NRR with CO₂RR is to highlight that the unusual potential-dependent NRR activity is unexplained by the conventional DFT calculations for the theoretical limiting potential. The discussion of CO₂RR vs. NRR is now revised as below in the Introduction.

“The CO₂RR activity increases with more negative potential and the maximum CO₂RR activity is observed at around -0.7 V (Fig. 1b). At $U = -0.7$ V which is more negative than U_L (-0.32 V), CO₂RR can be sufficiently facilitated and begins to decrease due to the mass-transfer limit. Thus, the potential-dependent CO₂RR activity can be qualitatively explained by conventional DFT calculations. NRR activity also increases with more negative potential at first, however, it begins to decrease quickly at -0.4 V (pH = 7.2) or -0.05 V (pH = 13) (Fig. 1b), before reaching its U_L (-1.29 V). This result indicates that NRR activity prematurely decreases with increasing reduction potential while its kinetics has not reached its expected theoretically maximum. NRR shows an unusual potential-dependent behavior which is unexplained by the conventional DFT calculations. Thus the premature decrease in the NRR activity should be attributed to the intrinsic properties of catalysts.”

18. The authors use the term coverage (θ) for N₂ and H in the micro-kinetic modeling section. Also, in Figure 4, the y-axis represents a continuously varying coverage. However, several systems tested in this manuscript are single-site catalyst (for example Fe@N₄). How is coverage defined for such a system?

Reply: The coverage is defined as the number of occupied Fe sites over the number of all Fe

sites. Despite the catalyst being a single atom catalyst, the catalyst as a whole contains many single atom sites, thus we can define coverage. This definition has been used for other microkinetic models.

19. How is pH incorporated when making the free energy surfaces? It is very clear that the bulk pH and local pH are very different. How do the authors treat the local pH? Also, based on my understanding the competition of NRR with HER should be more rigorous under acidic conditions provided the abundance of H^+ to block the active sites. However, this should be less severe under alkaline conditions.

Reply: The pH is incorporated by changing the chemical potential of H^+ (bulk) or OH^- (bulk), which are the reference state of the free energy surfaces. Since we used RHE scale, pH effect on reaction energy is naturally incorporated. We agree with the reviewer that the bulk pH and local pH are different due to the accumulated ionic species (e.g. OH^-) at the interface. To fully investigate the effect of local pH on energetics, pH should be explicitly considered in the DFT calculations. To the best of our knowledge, such a pH effect has not been rigorously assessed due to the computational cost in large-scale explicit simulations. Instead, Goddard and coworkers [9], Chan and coworkers [10], and Liu and coworkers [11] assessed the pH effect by considering the change in the activity of ions (the method used in our study) and were able to reproduce the experimental trend. Again, we agree that local pH as well as pH-dependent quantitative comparison of NRR vs. HER is an important question, but due to the computational challenge we believe that a dedicated separate investigation is necessary to fully address this topic. We added a brief discussion on the limitation of the pH effect treated in our study in p.15 of the main text.

[9] Mechanistic Explanation of the pH Dependence and Onset Potentials for Hydrocarbon Products from Electrochemical Reduction of CO on Cu (111). *J. Am. Chem. Soc.* 2016, 138, 483

[10] pH Effects on Hydrogen Evolution and Oxidation over Pt(111): Insights from First-Principles. *ACS Catal.* 2019, 9, 6194

[11] Enhancing the Understanding of Hydrogen Evolution and Oxidation Reactions on Pt(111) through Ab Initio Simulation of Electrode/Electrolyte Kinetics. *J. Am. Chem. Soc.* 2020, 142, 4985

20. The CHE definition in Supplementary Note 2 seems problematic. It is not the chemical potential of H^+ , but rather of the (H^+ and e^-) pair that can be related to the $\mu(H_2)$ at certain conditions. Also, credits should be given to the correct reference for CHE (i.e., Norskov 2004).

Reply: Chemical potential of proton can be related to the that of H_2 as $\mu(H^+) = \frac{1}{2}\mu(H_2) + \Phi_{SHE} - 0.059pH$. We omitted the derivation of $\mu(H^+)$ from the CHE model in original submission. The $\mu(H^+)$ can be derived from the CHE model as below.

The 0 V (vs. SHE) is defined when the 1 bar of gas H_2 is equilibrium with the pair of proton and electron at pH = 0 (i.e. $1/2 H_2 \leftrightarrow H^+ + e^-$). At 0 V (vs. SHE), the chemical potential of a pair of proton and electron is equal to half of the chemical potential of H_2 (**1**), which is the well-known CHE model.

$$\mu(H^+ + e^-) = \frac{1}{2}\mu(H_2) \text{ (1)}$$

The chemical potential of proton is decoupled from the equation (**1**).

$$\mu(\text{H}^+) = \frac{1}{2}\mu(\text{H}_2) - \mu(\text{e}^-) \quad (2)$$

From the definition of electrode potential (U) as in equation (3), $\mu(\text{e}^-)$ is equal to the $-\Phi_{\text{SHE}}$ at 0 V (vs. SHE). Here, Φ_{SHE} represents the absolute potential of SHE.

$$U \text{ (V vs. SHE)} = \frac{-\mu(\text{e}^-) - \Phi_{\text{SHE}}}{e} \quad (3)$$

Consequently, the $\mu(\text{H}^+)$ can be related to $\mu(\text{H}_2)$ as equation (4) by replacing $\mu(\text{e}^-)$ to $-\Phi_{\text{SHE}}$ in (2).

$$\mu(\text{H}^+) = \frac{1}{2}\mu(\text{H}_2) + \Phi_{\text{SHE}} \quad (4)$$

When $\text{pH} \neq 0$, the effect of pH on $\mu(\text{H}^+)$ is included as (5).

$$\mu(\text{H}^+) = \frac{1}{2}\mu(\text{H}_2) + \Phi_{\text{SHE}} - 0.059\text{pH} \quad (5)$$

We now added the above discussion in Supplementary Note 2 and cited the original paper for the CHE model.

Reviewer #3 (Remarks to the Author):

The authors have written a very nice manuscript and should be commended for producing a very clean, concise, and well-reasoned work of science. As I outline below, the results of this paper are somewhat expected. The main contributions of this paper are 1. The insufficiency of the CHE model in describing the potential dependence of the species relevant to NRR and 2. Per microkinetic modeling too much importance is put on the limiting potentials from DFT. This work clearly outlines and corrects a flaw in the current thinking and literature around NRR electrocatalysis by testing the CHE model and introducing microkinetic modeling. I thus recommend this manuscript for publication with minor revisions.

Reply: We are very grateful to the reviewer for positive view on our work. We respond to each question the reviewer brings one by one as below. The replies in this letter are marked in blue, while the revised texts in the manuscript and supporting information are highlighted in yellow.

Specific Comments:

1. The authors discuss the general trend of charge transfer being larger for H* adsorption than N₂* adsorption. While the demonstration of this trend is valuable, the trend itself is not surprising. N₂ is known to be very inert, and thus typically physisorbs on surfaces and thus has very limited charge transfer. Additionally, the greater dependence of H* on applied potential relative to N₂* is also expected. The adsorption energies of N₂* and H* are typically modeled using the computational hydrogen electrode model. In this model the adsorption of N₂ is assumed to be independent of applied potential, necessitating a crossover between H* and N₂* adsorption energies at some applied potential. The authors find this crossover as expected. While authors show in their manuscript that the CHE model is not sufficient to describe the real physics I present it here just to explain the expectations a reader would have.

However, the differing potential dependence of N_2H^* relative to H^* as well as N_2^* having significant potential dependence at all is interesting and not expected. Much of this lengthy discussion is outlined in [1] below (reference 56 in the manuscript.) I am somewhat surprised this work is not referenced more extensively, as this piece cleanly picks up the thread of this previous work.

Reply: We thank the reviewer for the insightful comment as well as pointing to an important reference. Adding the reviewer's insightful opinion, we also find that the potential dependence of $*H$ -forming activation process is also different for different mechanisms, namely Volmer (0.96) vs. Heyrovsky (0.56). This also leads to the lack of potential-dependent crossover in $*H$ coverages in the CHE model. We now added these discussion in the main text, and the reference [1] is now more extensively introduced in the Introduction as suggested by the reviewer.

“The possible reason for the premature decrease of NRR activity has been suggested qualitatively by the dominant H coverage at negative electrode potential since the H binding increases faster than N_2 binding by the electrode potential^{7,26,27}. Nørskov and coworkers suggested that the surface will be covered by hydrogen rather than N_2 at negative enough potentials by using reaction equations²⁶. However, such a possibility has not been theoretically verified. To understand the premature decrease of NRR activity, a comprehensive understanding of the potential-dependent competition of NRR vs. HER should be investigated based on the quantitative change of the coverage and kinetics.”

2. The authors examine the three surface species that are important for NRR: N_2^* , N_2H^* , and H^* . However, there is a 4th important species: NH_2^* . NH_2^* is often seen control the other half of the volcano plot when scaling relations are generated [2]. Thus, the authors cannot

claim to have fully examined the species relevant for NRR. This limitation should be noted.

Reply: We thank the reviewer for this comment. Catalysts which lie on the left-leg of the NRR volcano plot (i.e. strongly N binding), the reaction $*\text{NH}_2 + (\text{H}^+ + \text{e}^-) \rightarrow * + \text{NH}_3$ has been known as the potential-determining step. For these catalysts, as the reviewer correctly points out, $*\text{NH}_2$ rather than $*\text{NNH}$ should be considered. We now added this discussion in the “Origin and descriptor for different slopes” section (in Page 21) as below.

“Here, we estimate the overall kinetics of NRR by using $*\text{NNH}$ formation which is the PDS of Fe@N_4 as well as various catalysts. However, catalysts with strongly N binding strength, which lies on the left-leg of the NRR volcano plot, are limited by $*\text{NH}_2 + (\text{H}^+ + \text{e}^-) \rightarrow \text{NH}_3$ ^{6,62}. Thus, we note the potential-dependent energetics and charge transfer associated with $*\text{NH}_2$ should be considered for strongly N binding catalysts.”

3. The section titled “Origin and descriptor for different slopes” is an interesting discussion of why there are differing slopes in the adsorption vs potential of various species with respect to their extent of charge transfer. However, as the section lays out this is more a demonstration of the Nernst equation than a new discovery. This is not really a flaw, however it should not be posed as a major discovery.

[1] Singh, A. R. et al. Electrochemical Ammonia Synthesis-The Selectivity Challenge. ACS Catal. 7, 706-709 (2017).

[2] Computational Screening of Rutile Oxides for Electrochemical Ammonia Formation. (n.d.). doi:10.1021/acssuschemeng.7b02379.s001

Reply: We agree with the reviewer that the demonstration of the Nernst equation is not a new discovery. We now moved the Fig. 6, demonstration of the Nernst equation to Supplementary

Fig. 18. In this section, we now more focus on the different charge transfer in adsorbate rather than verification of relationship between charge transfer and the slope. Thus, the new Fig. 7 which compares the electron transfer of each adsorbate is now added in the manuscript.

Fig. 7 | The amount of electron transfer (ΔN_e) in $*H$, $*N_2$ and $*NNH$ formation at 0 V. Black, red, and blue colors represent $*H$ formation, $*N_2$ formation and $*NNH$ formation from $*N_2$, respectively.

REVIEWERS' COMMENTS

Reviewer #1 (Remarks to the Author):

In the revised manuscript, the authors have fully addressed the concerns as the reviewer raised. I have no more comments.

Reviewer #4 (Remarks to the Author):

In this work, Choi et al use DFT to analyze the competition between HER and NRR on a single-atom FeN₄ catalyst. They demonstrate that it is essential to consider the explicit charge state of the catalyst site using a grand canonical formalism to establish a constant electrode potential (CEP), which they compare to the computational hydrogen electrode (CHE) model. The CEP model captures non-integer charge transfer effects impacting the coverage dependence of H* and N₂* on potential (i.e., capturing deviations from predictions made with the simpler CHE model). The balance between H* and N₂* coverage varies with potential, which explains (in part) the experimentally observed competition between HER and N₂R. I see that the manuscript has already been reviewed and extensively revised. I believe that the authors have thoroughly answered the questions and comments raised by the other reviewers. For this reason, this manuscript should be accepted once the following minor points are considered.

1) The authors state on Pg. 3 that "Theoretical studies have suggested that the theoretical limiting potential (UL) where the electrochemical reaction becomes exothermic..." This phrasing is vague and should be restated to be more precise. UL is the potential at which all elementary reaction steps become energetically downhill (as opposed to the potential where the overall reaction becoming exothermic).

2) The authors state that "The lowest energy pathway based on the apparent activation energy is represented by red line in Fig. 3." How is the "apparent activation energy" defined in this case? E.g., is it defined as it is in the energetic-span model of Kozuch and Shaik (Acc. Chem. Res. 2011, 44, 2, 101–110)? The authors use the term "apparent activation energy" several times without defining what they mean by it. Many different definitions for "apparent activation energy" have been used in the literature, so this is a point that needs to be clarified.

3) Fig. 3 is difficult to read and interpret because many lines are overlapping, and it is not clear which label corresponds to which line. This figure should be revised.

4) The authors state in the rebuttal letter in response to comment 7 of reviewer 2 that "we included the activation energies of all the other reaction steps in the MKM." In Fig. 3 and Table S2 the authors report that some steps "proceed without a barrier." I am surprised to see that so many reaction steps (seven out of fourteen) have no barrier. How did the authors determine that these steps have no barrier?

5) In the revised text of the SI (Ln. 86) the authors state that "Under acidic conditions, only the *NNH formation, the RDS of NRR under negative potential, is included in calculating activation energy. Except for the *NNH formation, reaction energies were calculated without explicit water layer." This phrasing is unclear. Do the authors mean that, for the case of acidic conditions, only the *NNH formation barrier was calculated explicitly and all other barriers were neglected? If so, the fact that explicit barriers for all steps were considered for the neutral/alkaline cases but not for the acidic case should be stated more clearly (and justified more clearly). Also, Why did the authors assume that the explicit solvation layer was important for the *NNH and not for the others? How much should we expect the energetics to change with and without explicit solvation?

Reviewer #4 (Remarks to the Author):

In this work, Choi et al use DFT to analyze the competition between HER and NRR on a single-atom FeN₄ catalyst. They demonstrate that it is essential to consider the explicit charge state of the catalyst site using a grand canonical formalism to establish a constant electrode potential (CEP), which they compare to the computational hydrogen electrode (CHE) model. The CEP model captures non-integer charge transfer effects impacting the coverage dependence of H* and N₂* on potential (i.e., capturing deviations from predictions made with the simpler CHE model). The balance between H* and N₂* coverage varies with potential, which explains (in part) the experimentally observed competition between HER and N₂R. I see that the manuscript has already been reviewed and extensively revised. I believe that the authors have thoroughly answered the questions and comments raised by the other reviewers. For this reason, this manuscript should be accepted once the following minor points are considered.

Reply: We are very grateful to the reviewer for the positive review on our work. We respond to each question of the reviewer one-by-one below. The replies in this letter are marked in blue, while the revised texts in the manuscript and supporting information are highlighted in yellow.

1) The authors state on Pg. 3 that “Theoretical studies have suggested that the theoretical limiting potential (UL) where the electrochemical reaction becomes exothermic...” This phrasing is vague and should be restated to be more precise. UL is the potential at which all elementary reaction steps become energetically downhill (as opposed to the potential where the overall reaction becoming exothermic).

Reply: We thank the reviewer for this correction. We now edited the sentence to “Theoretical

studies have suggested that the theoretical limiting potential (U_L) where the all electrochemical elementary steps become exothermic...

2) The authors state that “The lowest energy pathway based on the apparent activation energy is represented by red line in Fig. 3.” How is the “apparent activation energy” defined in this case? E.g., is it defined as it is in the energetic-span model of Kozuch and Shaik (Acc. Chem. Res. 2011, 44, 2, 101–110)? The authors use the term "apparent activation energy" several times without defining what they mean by it. Many different definitions for "apparent activation energy" have been used in the literature, so this is a point that needs to be clarified.

Reply: We thank the reviewer for this comment and agree that there are many different definitions for the apparent activation energy. Here, the apparent activation energy is defined as the energy difference between the highest transition state (TS) energy and the lowest energy intermediate in the catalytic cycle from the energetic span model. We now added such a sentence with the reference suggested by the reviewer in Page 9 and clarified the definition of apparent activation energy in this study.

3) Fig. 3 is difficult to read and interpret because many lines are overlapping, and it is not clear which label corresponds to which line. This figure should be revised.

Reply: We appreciate the reviewer’s suggestion. We changed the color of a black line to gold when three lines overlap and clarified which label corresponds to which line.

4) The authors state in the rebuttal letter in response to comment 7 of reviewer 2 that “we included the activation energies of all the other reaction steps in the MKM.” In Fig. 3 and Table S2 the authors report that some steps “proceed without a barrier.” I am surprised to see

that so many reaction steps (seven out of fourteen) have no barrier. How did the authors determine that these steps have no barrier?

Reply: We did find the transition states (TS) for all proton-coupled electron transfer (PCET) steps. However, activation energy becomes negative for several reactions after the OH⁻ correction. The accurate calculation of solvation free energy of OH⁻ under the periodic boundary conditions is still challenging. For example, the calculated solvation free energy of OH⁻ by the implicit solvation model implemented in the VASPsol (-2.294 eV) is more endothermic than experimental value (-4.553 eV) (*J. Chem. Theory Comput.* 2016, 12, 1331). To correct the solvation free energy of OH⁻, several approaches have been reported. Here, we corrected the energetics of OH⁻ species following the Liu et al (*J. Am. Chem. Soc.* 2020, 142, 14985) and add -0.702 eV for OH⁻ species in the TS throughout this study. After the OH⁻ correction, activation energy of several highly exothermic reactions (e.g. $\Delta G < -0.4$ eV even at 0 V) becomes negative (disappears) which we denote as a barrierless step. We now added such discussions in the Supplementary Note 2 and revised Supplementary Table 2.

5) In the revised text of the SI (Ln. 86) the authors state that “Under acidic conditions, only the *NNH formation, the RDS of NRR under negative potential, is included in calculating activation energy. Except for the *NNH formation, reaction energies were calculated without explicit water layer.” This phrasing is unclear. Do the authors mean that, for the case of acidic conditions, only the *NNH formation barrier was calculated explicitly and all other barriers were neglected? If so, the fact that explicit barriers for all steps were considered for the neutral/alkaline cases but not for the acidic case should be stated more clearly (and justified more clearly). Also, Why did the authors assume that the explicit solvation layer was

important for the *NNH and not for the others? How much should we expect the energetics to change with and without explicit solvation?

Reply: In neutral or alkaline conditions where experiments have been reported, we rigorously calculated the activation energy of all PCET steps and compared the results with the experiments. However, since the experiment in acidic condition has not been reported, to simplify the treatment we only calculated the $G_a(*N_2 \rightarrow *NNH)$, the RDS of NRR, to verify the premature decrease of NRR activity in acidic condition. We find that the MKM results do not change significantly whether all G_a are included or only the $G_a(*N_2 \rightarrow *NNH)$ is included. As considering only the $G_a(*N_2 \rightarrow *NNH)$ is sufficient to estimate the NRR activity, we calculated ΔG without explicit water layer for other elementary steps of NRR to save the computational cost. We find that the mean absolute difference of ΔG with and without explicit water layer is 0.10 eV, which is not expected to affect the overall NRR kinetics significantly. We now added such discussion and Supplementary Fig. 8 in Supplementary Note 2.

Supplementary Fig. 8 | Effect of $G_a(*N_2 \rightarrow *NNH)$ and explicit water layer. (a) NH_3 yield

rate obtained by MKM simulations under alkaline conditions ($\text{pH} = 13$). Black and red lines represent MKM result including all Ga included and that including only $G_a(*\text{N}_2 \rightarrow *\text{NNH})$. (b) Comparison of ΔG of NRR intermediates with and without explicit water layer at 0 V vs. RHE at $\text{pH} = 13$.